# Simultaneous Localization and Mapping (SLAM) and Data Fusion in Unmanned Aerial Vehicles: Recent Advances and Challenges

**Abhishek Gupta** *,† and **Xavier Fernando** †

Department of Electrical, Computer and Biomedical Engineering, Ryerson University, Toronto, ON M5B2K3, Canada; fernando@ryerson.ca
* Correspondence: abhishek1.gupta@ryerson.ca
† These authors contributed equally to this work.

**Abstract:** This article presents a survey of simultaneous localization and mapping (SLAM) and data fusion techniques for object detection and environmental scene perception in unmanned aerial vehicles (UAVs). We critically evaluate some current SLAM implementations in robotics and autonomous vehicles and their applicability and scalability to UAVs. SLAM is envisioned as a potential technique for object detection and scene perception to enable UAV navigation through continuous state estimation. In this article, we bridge the gap between SLAM and data fusion in UAVs while also comprehensively surveying related object detection techniques such as visual odometry and aerial photogrammetry. We begin with an introduction to applications where UAV localization is necessary, followed by an analysis of multimodal sensor data fusion to fuse the information gathered from different sensors mounted on UAVs. We then discuss SLAM techniques such as Kalman filters and extended Kalman filters to address scene perception, mapping, and localization in UAVs. The findings are summarized to correlate prevalent and futuristic SLAM and data fusion for UAV navigation, and some avenues for further research are discussed.

**Keywords:** multimodal sensor fusion; sensor data fusion; simultaneous localization and mapping; unmanned aerial vehicles



## 1. Introduction

With the emergence of 5G wireless systems, mobile edge computing, and cloud networks, unmanned aerial vehicles (UAVs) have gained attention in diverse applications [1,2]. Also known as drones, or remotely piloted aircraft systems (RPAS), UAVs contribute to mobility transformation, and impact service delivery in many applications [3], such as e-commerce delivery, remote surveying, providing cellular and wireless coverage, etc. Dense urban environments, campuses, amusement parks, etc., can be easily accessed with a UAV to geo-reference miles of data [4,5]. Effective communication between UAVs and supporting infrastructure is critical to deliver these services, and this is partly accomplished using perception, planning and control. However, unlike a vehicle trajectory or a robot path, which are determined through pre-existing geospatial maps based on specified co-ordinate systems, a UAV path (trajectory) lacks a deterministic map [6,7]. As the co-ordinates of a UAV trajectory change rapidly and abruptly, a pre-determined 3D map of spatial coordinates through which a UAV traverses is not feasible, especially in the realms of mountainous terrains, hills, valleys, etc. [8]. Moreover, in applications such as smart cities, UAVs need to interact with multiple ground-based entities for tasks such as energy-harvesting, reliable ground-entity localization, in addition to communication [9,10]. The spatial co-ordinate system used to locate a UAV needs to be compatible with the co-ordinate system that locates the on-ground entity. However, having these co-ordinates in the same format may be computationally expensive [11,12].

UAV localization has been proposed using a multitude of technologies, including visible light communication (VLC). However, the deployment of light emitting diodes (LED) based transmitters in UAVs is still an active area of research [13,14]. Global navigation satellite systems (GNSS) are also used to measure the time of arrival (ToA) and received signal strength (RSSI) of satellite signals to estimate the location of the autonomous UAV [15]. Based on multiple satellites, GNSS receivers can estimate a target-position through multilateration [15]. However, in the context of UAVs, this approach has issues caused by variations in atmospheric conditions, signal interference, and inexpensive commercial global positioning system (GPS) receivers causing poor accuracy. Moreover, GNSS receivers are affected by non-line-of-sight (NLoS) signal reception and are generally more accurate for longer distances compared to estimation over short-distances [16,17]. By adding specialized hardware infrastructure, such as differential GPS (DGPS) and real-time kinematics-GPS (RTK-GPS), these errors can be reduced to some extent. Utilizing mmWave for localization also yields better accuracy [18,19]. As GPS precision is in the order of a few centimeters, it is an active area of research to capture objects larger than an inch in UAV based mobile mapping systems [20], as well as to correlate mapping with UAV localization using GPS [21].

Rather than relying on GPS or VLC for localization, a promising technique is simultaneous localization and mapping (SLAM) that uses environment mapping for real-time UAV localization [22,23]. Unlike conventional mapping and localization, known as dead reckoning, UAV mapping uses raw sensor data that is categorized based on the landmarks it represents in order to provide visual reference and state estimation [24]. Therefore, SLAM is envisioned as a potential technique to keep track of UAV positions in real-time in GNSS-denied or GNSS-degraded environments [17]. Humans may interact with external surroundings and simultaneously acquire information from sound, visuals, and other sensory inputs as well. Perception gives meaning to these sensory inputs, allowing the information to be deciphered, and appropriate actions to be taken. For example, if a perceived sound is that of a fire alarm, the action is to evacuate the building [25]. Similarly, safe and optimal trajectory planning depends on environment perception and precise UAV localization [26]. Perception relies on the types of sensors in the UAV and a constant stream of data from sensors that is translated into meaningful information. Note that selecting appropriate sensors for different weather conditions, traffic, applications, etc., will also impact overall accuracy [25]. Many low-cost, off-the-shelf, consumer-grade sensors may also be deployed in UAVs with sensor data fusion [27].

This paper explores data fusion for simultaneous localization and mapping in UAVs using classical approaches, and analyzes some open problems in scene perception for UAVs using SLAM and sensor data fusion. We answer the following questions pertaining to the contribution of SLAM and data fusion in UAVs.

1. What are the fundamental operational requirements for fully functional UAVs?
2. What developments have been achieved in UAV localization in the last 10 years and what are some promising research directions for the next decade?
3. How does SLAM achieve perception in UAVs? Is it feasible to attain human level cognition and perception in UAVs using SLAM?
4. What are the most recent SLAM techniques applied to UAVs and promising directions for further research?
5. Why is data fusion a promising technique for solving object detection and scene perception in UAVs?
6. Which sensors are used for object detection and scene perception in UAVs and how is multi-sensor data fusion and 3D point cloud analysis realized ?

The remainder of this paper is structured as follows. Section 1 introduced the context of this work and highlights the theme of the paper. This sections outlines the central questions pertaining to data fusion and SLAM in UAVs, that have been answered in the subsequent sections. Section 2 discusses UAV applications and analyses why SLAM is a critical requirement in many of these applications for safe and reliable UAV navigation.

Section 3 explores scene perception in UAVs, commonly used sensors, and techniques to perform multimodal sensor fusion. This section highlights the challenges of concatenating the different nature and format of data gathered by various sensors to extract meaningful information in the context of UAV path planning and navigation. In Section 4, the attributes and building blocks of SLAM, namely Kalman filters (KF), extended Kalman filters (EKF), and particle filters, are explored. The section also discusses the state-of-the-art Bayesian and probabilistic approaches for sensor data fusion and SLAM. Section 5 describes visual SLAM and UAV localization based on image registration, where distinction between static and dynamic obstacles increases localization accuracy. This section also discusses visual odometry (VO) to localize a UAV in a previously mapped region by state estimation and drift induced in the photogrammetry based mapping process. Section 6 discusses some open issues that impact the ongoing trends in the field and proposes some avenues for future work. Lastly, Section 7 concludes the survey.

## 2. UAV Applications

A few applications and use cases of UAVs are depicted in Figure 1 and are briefly mentioned below:

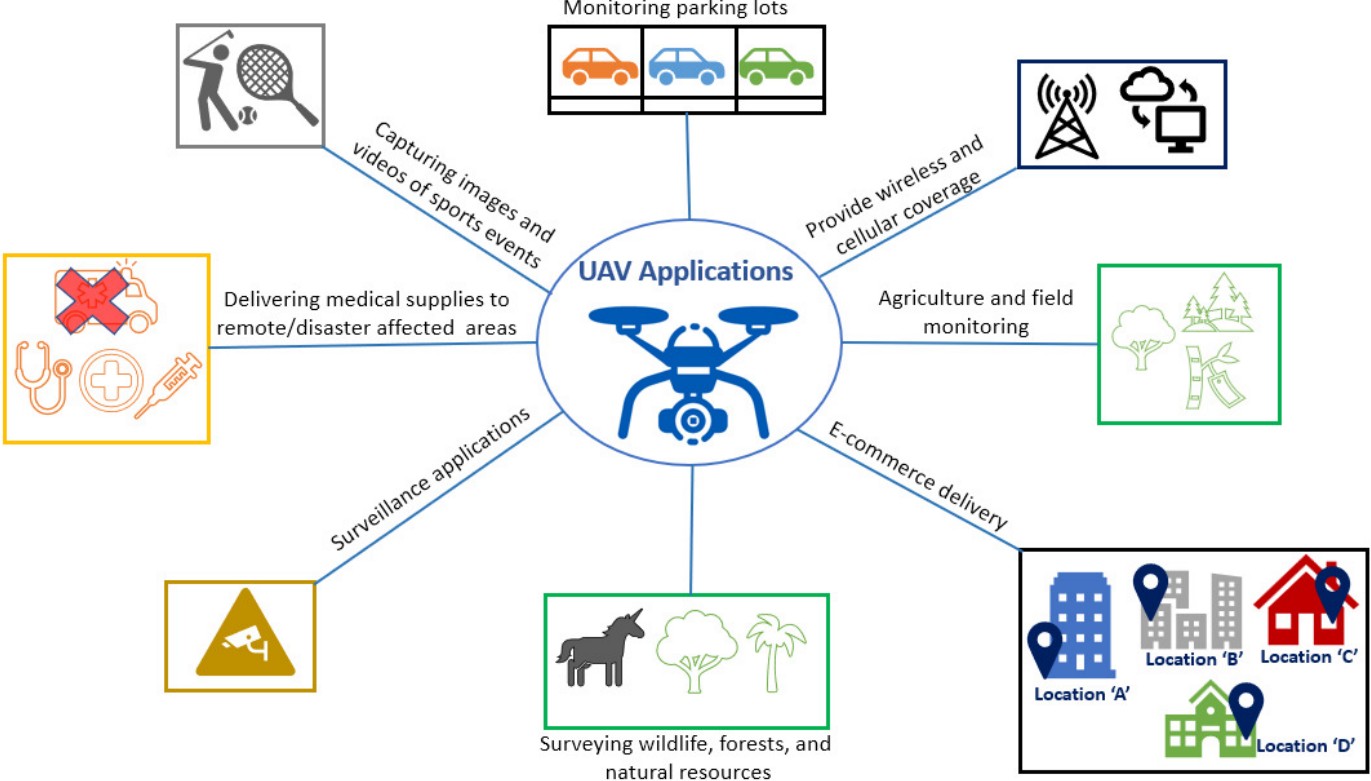

**Figure 1.** A few applications of UAVs.

- UAVs can be used to deliver e-commerce packages, medical equipment, medicine, food and medication to remote or disaster affected areas [28,29], as long as the weather is good and there is at least a small space to land [8].
- In agriculture and farming, UAVs enable field surveillance that allows farmers to remotely monitor crops and vegetation ready for harvest and damaged by pest infestation, and check frost levels in the fields [30]. UAVs can also be used for spraying pesticides and are considered safer and more precise than manual spraying. Fruit farms and orchard management with UAV image-processing yield better outcomes [31].
- UAVs are used to survey forests, wildlife, natural resources, and measure air pollution.

- UAVs also find widespread applications in structural inspection, architectural surveying and mapping, aerial photography, content-based remote-sensing, image retrieval, and image localization [32–34]. Inspection of highway infrastructure and scheduling of repairs have also been accomplished with UAVs [35].
- As UAVs can carry cameras to any difficult-to-reach region, UAVs find widespread applications in search and rescue operations [36].
- Flying UAVs above a recreation or sports event allows dynamic shots to be captured from varied angles that would otherwise not be possible [37,38].
- UAVs can play a vital role in the military for warfare purposes and border surveillance [39]. Some military-grade UAVs can remain airborne for weeks and span approximately a million kilometers before recharging the batteries [39]. Such UAVs can also assist fire fighters to locate hot spots that indicate fires and transmit a live video via Wi-Fi or cellular networks.
- When embedded with miniaturized antennas and RF transceivers, UAVs are useful to enhance wireless network coverage in areas of poor coverage where installing communication towers is not feasible [40]. This feature is especially beneficial in remote areas or areas affected by flood, earthquake or other disasters, as UAV based radio access networks can be rapidly deployed in an ad-hoc manner [3].

Most of these applications require autonomous UAV navigation and precise localization—an accurate map of the environment and information about where a UAV is at a given time [41]. The current limitations, expectations and challenges in UAV data fusion and SLAM are enlisted in Table 1. UAV environment is usually represented by using maps that capture static and mobile landmark information such as buildings, roads, pedestrians and vehicle traffic. UAVs also need to be aware of other UAVs in the 3D vicinity to avoid collisions. Visual recognition systems (VRS) are embedded in most UAVs for image classification, object detection, segmentation, and localization for basic ocular performance and to actuate kinematic manoeuvres [42,43]. An effective navigation system enables a UAV to know where the target landmark is and prevents it from wandering in a haphazard manner and hitting an object [44]. Due to the dynamically varying nature of a UAV path, the reliable perception of the environment and precise localization becomes a challenging task. This task is accomplished by various sensors installed in a UAV to:

1. Know where a UAV is at a given time $t$, defined as localization [26]. The localization problem can be resolved to some extent through short-term maps and trajectory computed using VO [16]. For instance, a radio localization device installed on the UAV can be paired with a GNSS receiver to provide a short-term trajectory and map.
2. Estimate the surrounding environment in terms of co-ordinates and images, defined as map building. In SLAM, map-building is used for perception, planning, and control [45], briefly described as follows:

   - Perception is the ability of a UAV to discern meaningful information from its sensors to understand the environment [46]. Both localization and map building enhance a UAV's perception. As an autonomous entity, a UAV needs to understand its own state, location, the external environment, and the map. Perception leads to safe UAV path planning [47].
   - Planning involves making decisions to achieve the trajectory objectives [48].
   - Control refers to the ability of a UAV to execute the planned actions. Due to accumulation of errors, the accuracy of a map degrades with time. Visual odometry is used to optimize the trajectory and map over a longer timestep for accuracy and performance. In SLAM, the localization and mapping tasks are dependent [47].

**Table 1.** Data fusion and SLAM in UAVs: Current limitations, expectations & challenges.

| Relevant Aspect | Currently | Expected with SLAM |
|---|---|---|
| GNSS vs. SLAM requirement | GNSS signal strength impacts localization. | Automatic localization in dynamic scenarios [26]. |
| | Low capabilities for path planning and UAV trajectory estimation. | Enhanced path planning and UAV trajectory estimation across dynamic terrains [44,47]. |
| | Costly to provide wireless communications and networking services. | SLAM and data fusion can facilitate UAV network services in an ad hoc manner [2,3]. |
| | Application scenarios lack robustness. | Robust UAV path planning in disaster relief missions [28,29]. |
| | GNSS coverage interrupted in rural zones. | SLAM and data fusion make use of computer vision [40,49]. |
| | Computing resources needed for ground target localization. | Dynamic localization and resource provisioning in response to sensor data [50,51]. |
| UAV Applications | GNSS may not be available in natural disasters and response scenarios. | With SLAM, only targeted locations are considered for UAV trajectory [8]. |
| | GNSS is suitable for remote proximity monitoring between fixed target locations. | Data fusion and SLAM support proximity monitoring between fixed and mobile targets using sensor-mounted UAVs [34]. |
| | Target landmark scaling is limited. | SLAM offers incremental scaling of target locations. |
| | Visual tracking of target landmarks is dependent on signal strength. | Diverse due to use of visual odometry, photogrammetry, and sensor data sequence [32,35]. |
| Sensing and Analysis Approaches | Data from all landmarks is collected. | Data from only the targeted landmarks is collected [31]. |
| | Massive bandwidth required for robust cooperative positioning of UAVs. | Efficient use of bandwidth through robust cooperative UAV positioning [52]. |
| | End-to-end scene perception. | Feasible in cluttered GPS-denied environment [53–55]. |
| | Depends on spatial orientation of target. | Low-cost sensors useful in GPS-denied environment [23,24]. |
| Multi-UAV trajectory planning Services | Complex task allocation and data fusion. | Simplified collaborative SLAM [56–58]. |
| | Limited coordination in varying spatial configuration. | Seamless and cognitive neighbour-aware decision making [59,60]. |
| | Throughput maximization needs wireless power. | SLAM and data fusion optimize edge computing through synergy between vision and communications [61,62]. |
| | Filter based UAV pose tracking. | Refined and non-linear pose tracking [63]. |
| | Precise positioning is difficult. | Precision under uncertainty and is error aware [64]. |
| | Low accuracy photogrammetry. | Dynamic and enhanced visual SLAM [65]. |
| | Data from UAVs hard to integrate. | SLAM facilitates simplified sensor data integration [66]. |
| 3D Characterization | Poor in low-texture environment. | Better due to visual-inertial SLAM and sensor fusion [67,68]. |
| | Camera-based target tracking. | Spatio-temporal observations from multiple sensors [20,69]. |

**Table 1.** *Cont.*

| Relevant Aspect | Currently | Expected with SLAM. |
|---|---|---|
| UAV Deployment & Management | UAV swarms hard to track. | Smart fusion of multi-sensor data for UAV localization [17]. |
| | UAV placement a priori in large paths. | Dynamic placement of UAVs for any trajectory [19,21] |
| | Integrated sensor fusion with GNSS. | SLAM and data fusion geo-reference target landmarks, conserve computing resources using 3D point-clouds [4,5]. |
| Other Issues | Energy saving strategies are limited. | Flexible energy saving strategies [70]. |
| | Complex and static control techniques. | Flexible trajectory control techniques with enhanced FoV [71] |

## 3. Multimodal Sensor Fusion

Concatenating data from a multitude of sensors and estimating dynamic environments is referred to as multimodal sensor fusion [72]. Due to inherent limitations, the sensor data may have uncertainty which is handled using Kalman or extended Kalman filters, as well as sequential Monte Carlo techniques to achieve the best estimate [73]. The UAV trajectory and the environment map are referred to as the state [74]. When a set of longitude–latitude pairs describe the altitude of a UAV corresponding to the geographical coordinates, noise ($\mathcal{N}$) may be carried forward from one timestep to the next, degrading the accuracy of the current estimation [75]. In order to use noisy input to estimate the state of a UAV under uncertainty, evidential belief reasoning techniques such as the Dempster–Shafer theory are used to assign beliefs to the data, as well as to the combination rules to fuse the sensor data [76]. The Dempster–Shafer theory addresses conflicting and missing information by improving the mapping of the dynamic environment that represents uncertainty in multiple dimensions called the frame-of-discernment (FoD) [77]. For data fusion with constraints, multi-sensor fusion inference from each sensor is subject to uncertainty. Another technique known as fuzzy logic assigns a real number $\mathcal{R} \in \{0,1\}$ to signify the degree of truth or significance of the data [78].

In UAV mapping systems, the aerial data is translated into a coordinate system, and the captured landmark data is georeferenced and assigned to maps [79]. The resulting sensor data may be in the form of a point cloud, which is a dense collection of data points taken from a LiDAR sensor to create a precise 3D representation of the landmarks in the Cartesian coordinate system. Point cloud association by matching two independent scans, also known as scan matching or point matching, creates a globally consistent map [80]. Iterative closest point (ICP) is used in point-to-point and feature-to-feature matching methods to discover a linear transformation $\mathcal{T} = [\Delta s, \Delta \theta]$ that aligns two point clouds and estimates the UAV localization [81]. Note that sometimes it is possible that the point clouds from two scans are incorrectly matched to the same environment [82]. Hence, to improve matching characteristics, a matching framework based on features such as frame-to-frame measurements or maximum curvature points are used to sample the 3D landmark scans for mapping [70].

UAV sensors are non-invasive and capture short and long range data with higher point-cloud density and data fusion techniques. These are needed to maintain high precision while projecting data into other coordinate systems [37]. Geo-referencing data and control points use physical properties to capture variations in target features in the point cloud data [70]. Furthermore, sensor data fusion considers a number of factors:

- The nature and format of data collected by each sensor [83]
- Sensor's field of view [71]
- Synchronization times of various sensors [72]
- Data capture frequency [27]

- Sensor resolution and data packet size [27]
- Data association and calibration to correlate data from different sources [25]
- Most appropriate fusion approach [72]

Multisensor data fusion is a delicate task. Sensors do not always produce rich and dense data. Sometimes the data have redundancy that is compressed [84]. Fusion algorithms store numbers and matrices in a memory [78]. A good fusion algorithm takes advantage of the nature of the data that may also depend on the sensor characteristics. Note that SLAM is the process of estimating a UAV's starting location and gathering surrounding data as the UAV trajectory is updated in real-time [85]. Diverse representations of mapped data show consistent 3D landmark characteristics for localization and map fusion despite differences in sensor fields-of-view [71]. Multiple sensors create intermediate maps at each time step, and the UAV state is estimated by marginalizing past values [86].

As depicted in Figure 2, the data captured by the sensors will represent different parts of the UAV scene. This is further processed to obtain complete global information about the landmarks as well as the UAV trajectory [59]. Many sensors, such as RGB-D sensors, cameras, and other vision sensors capture data in the form of images. The 3D laser scanners and camera images are combined to predict object bounding boxes in the landmark data [87]. Moreover, for terrain classification such as grass, trees, buildings, vehicles, traffic lights, pedestrians, etc., the information related to the trajectory is shared among multiple sensors. However, the inherent statistical difference between multiple images or sensor data might lead to significant variations in reference maps and error in trajectories [88]. Temporal fusion generates a global map that represents the UAV navigation environment through a scan that corresponds to a set of points measured during one sensor rotation [89].

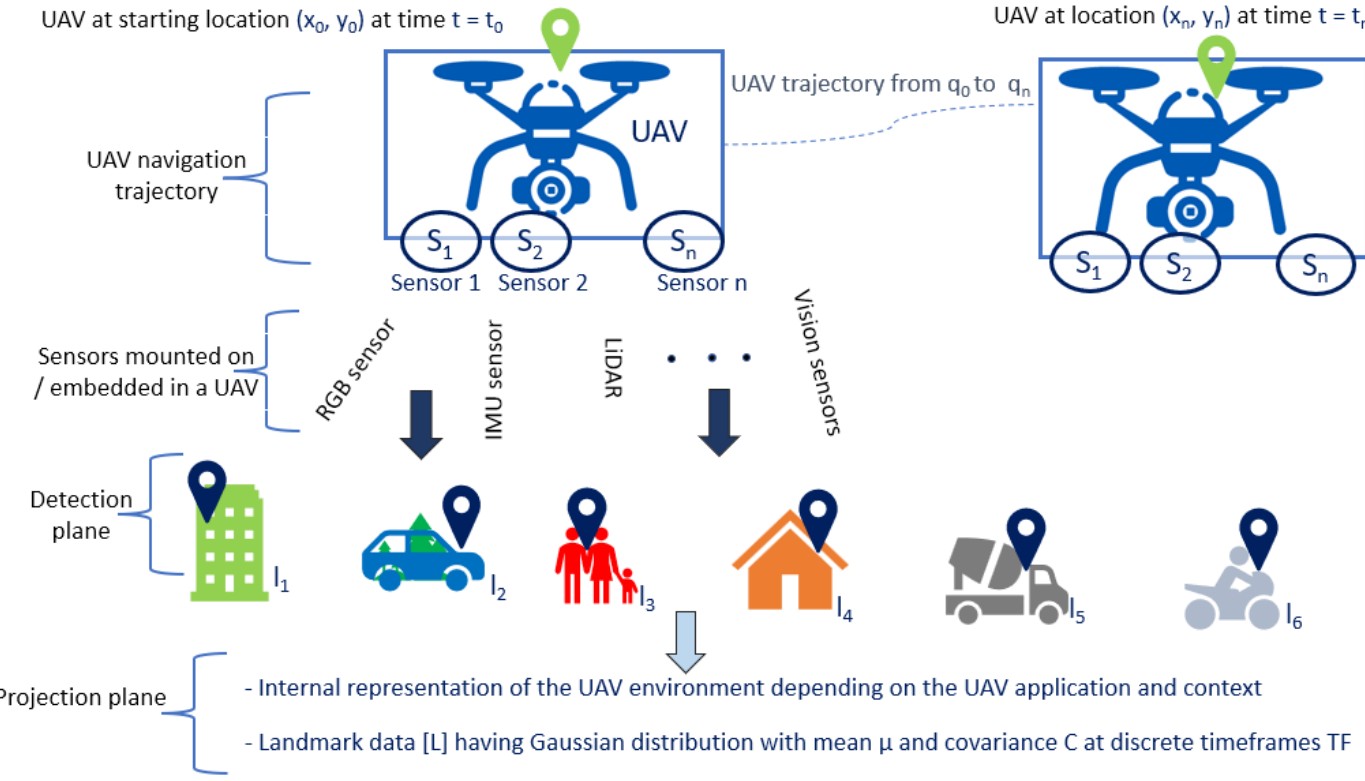

**Figure 2.** Depiction of a multimodal sensor data fusion approach for UAV localization and mapping to enhance environment perception with robust scene representation by capturing a global map of visited landmarks.

### 3.1. Challenges in Multimodal Sensor Fusion

Some challenges in multimodal sensor fusion for semantic scene understanding in UAVs are,

(i)　Impact of uncertainty and imprecise data: Each sensor operation is based on a particular data acquisition framework, which might differ based on the target phenomenon of interest. Therefore, the weight of each sensor outcome should be different. Furthermore, when merging data from different sensors, the fusion framework must reduce the impact of uncertainty and imprecise data for a broader representation of the environment [78].

(ii)　Different data formats: Raw sensor data, with a multitude of formats, are input to a fusion algorithm to obtain meaningful and relevant information. Therefore, the datasets need to be formatted to a universal file format that can be read by the fusion algorithm [90].

(iii)　Edge/Cloud processing: While cloud computing requires relaying the data to the cloud server for computing, the edge computing architecture enables the sensor data to be processed at the source of information. The choice between edge or cloud processing in UAV environments depends on latency, amount of data to be processed, available bandwidth, overall cost, and the feasibility to deploy edge servers in remote aerial locations. Using edge computing to locally process the captured sensor data while using cloud for overall SLAM analytics is a preferred approach [60].

(iv)　Collaboration and coordination among multiple UAVs: Signal unavailability or interference in an outdoor environment may corrupt the sensor data and induce errors [60]. Therefore, it maybe required to coordinate sensor data from multiple UAVs for optimum target coverage with low power consumption. This approach will reduce the possibility of missing target landmarks as well as prevent redundant coverage. Here, target coverage areas can be assigned to specific UAVs [61].

(v)　Noise and Bayesian inference: Inference methods such as Bayesian probabilistic fusion, evidential belief reasoning and fuzzy logic rely on probability distribution patterns to counter the effect of uncertainty in the sensor data. Sensor performance is affected by environmental factors such as changes in the magnetic field, temperature, etc., that add uncertainty to the measured data [76,91].

(vi)　Detecting overlap: Detecting overlap between individual sensor data can enhance complementary or cooperative sensor data fusion and reduce redundancy. Data from two or more sensors about the same target could be fused to enhance confidence in the data. For example, the data pertaining to overlapping landmarks in visual sensors are considered to be redundant while the data on the same landmark captured by two sensors with different fields of view are considered cooperative [92].

(vii)　Reliability issues: Often, the sensor data are not just uncertain, but could also be unreliable. One sensor may offset the (dis)advantage of another sensor. As one unreliable sensor may cause incorrect fusion results, reliability evaluation is indispensable in UAV SLAM applications [93].

### 3.2. UAV Data Fusion Requirements and Different Sensors Used in Practice

The objective of SLAM is to compute the position and orientation of UAVs relative to an initial reference position [94]. The low-cost consumer-grade sensors embedded in commercial UAVs have low precision that makes them susceptible to external factors. These sensors are often paired with other sensors for data fusion [67]. The sensor parameters may also vary according to the UAV's trajectory and velocity. As the UAV moves from one location to another, data fusion allows a global map of the UAV trajectory, as well as target landmarks, to be obtained without losing the previous mapped regions [95]. The dynamic updating of landmark information is crucial for SLAM algorithms so that the UAVs can perceive the challenging environment more vividly with enhanced environmental representation. Multi-sensor data fusion enhances mapping and environmental

representation with a large number of detection features [96]. Non-intrusive sensors such as laser scanners, cameras, wheel encoders, and inertial measurement units (IMUs) can detect movement in indoor environments where GPS localization does not work [97]. These sensors measure indirect physical quantities rather than direct location. For example, a wheel encoder measures the wheel rotation angle, while an IMU measures the angular velocity and acceleration. A camera or a laser scanner measures the external environment in specific ways [98]. However, mitigating the inaccuracy induced due to the distance between the camera and the target location is a challenging task [23].

A UAV cannot be assumed to navigate safely in free space; mobile objects and obstacles must be located in order to plan their paths safely. A UAV navigating in an unknown region must detect free space and when new information becomes available in an unknown area, it must be added to the fused data [53]. Different sensors, such as a 2D laser scanner and stereo cameras, embedded in a UAV capture and merge data while also detecting obstacles that might be hidden from the UAV [49]. The purpose of sensor fusion is to accurately detect target landmarks in all types of terrains [25]. In the event of an obstacle detection, some sensors cannot detect further as their field-of-view is blocked. Once two stereo images are received, image rectification projects the images onto one common image plane [99]. A dynamic environment is characterized by fast motion, vigorous rotation, and low texture. A change in view angle is used to obtain a deeper understanding of the scene's 3D structure [100]. For example, the fusion of LiDAR and RGB camera images enhances the mapping process to represent the environment [50]. The localization methods use image recognition to locate UAVs while simultaneously mapping the environment, perforingm sensor fusion, and perforingm VO to estimate the UAV's displacement [52]. In dynamic UAV environments, the external environment is subject to certain constraints where the accuracy of a localization algorithm is a function of whether those constraints are met in practice or not [51]. To achieve adaptability, robustness, and scalability, extensive parameter tuning is required for each sensor, and the number of parameters that need to be tuned in a given scenario dynamically varies [101]. For example, a static image from one place may be captured at a specific rate, while at another location, a continuous video stream might be captured at a relatively higher rate.

The depth or distance between objects cannot be obtained using a single image. This distance is critical for a UAV to perceive distances for scenes, establish a relationship among the objects, and correlate them with their approximate size [102]. The RBG cameras capture solid, non-reflective and non-shiny objects. Dark surfaces add noise to the RGB images and closer objects occlude distant objects. If a camera moves to the right, the objects in the image appear to move to the left. Moreover, closer objects move faster and distant objects move slower, leading to pixel disparity [69]. A big far away object, and a close small object may appear to be of the same size in an image due to the perspective projection effect [69]. Place recognition allows the UAV to detect previously visited locations and compute relative measurements [102]. As the camera scale factor leads to the same size images of different target landmarks, UAVs need to distinguish object sizes to determine the actual size of those objects to distinguish them [102]. Furthermore, in the presence of a shadow in an image, UAVs need to distinguish the distance and size of an object from a corresponding shadow. Recovering the 3D scene structures such as distance and size of the objects from constantly varying the camera's view angle and to simultaneously estimate UAV motion is simplified by using LiDAR point cloud processing [80,103].

Some commonly used sensors in UAVs along with their key characteristics are radars, inertial measurement units (IMU) and vision sensors. Typically, some of the commercial short-range radars operate at 24 GHz and detect obstacles up to 100–150 m away. Long-range radars operate at 77 GHz and detect obstacles as far away as 250 m in different environmental conditions. In addition, they generate less data and require less computing power [104]. An IMU sensor measures linear and angular motion with the help of gyroscopes and accelerometers [97]. Once connected to a UAV, they provide a continuous stream of data related to acceleration along the three principal axes and angular velocity

of the UAV. Errors along the $x$ and $y$ axes and how frequently data can be collected are limitations of IMUs [98].

A vision sensor detects objects by their texture, color and contrast. It can be used in UAVs to detect traffic lights, pedestrians, other vehicles, and lane markings. A perspective camera covers a 30°–45° horizontal field-of-view while some UAV applications demand that the cameras cover a wider range [105]. A camera may be of a monocular (only one camera), stereo (two cameras), RGB-D or fish-eye type. The RGB-D cameras can also measure the distance of the camera from the target landmark for each pixel [106]. In SLAM, multiple cameras are used, which are different from single-lens reflex (SLR) cameras that are more common in image processing [107].

(a) Monocular Camera: The monocular camera is simple, cost-effective, and easy to operate. The camera projects a 3D environment in a 2D form due to its 6 degrees of freedom (DoF) movement [77,89,108]. However, accuracy of the obtained map is based on the uncertainty associated with 6-DoF rigid body transformations [109]. The depth can be estimated from a single image but it leads to scale ambiguity as a result of translational movement. The collection of spatial points is calibrated with the intrinsic and extrinsic parameters. In a given set of points, point A and point B may be connected, while point B and point C may not be [67]. Once the distances are known, the 3D scene structure can be retrieved from a single frame to eliminate scale ambiguity. Monocular cameras estimate camera position, illumination changes, and scene structure using per-pixel depth estimates [110]. The depth estimate is achieved by comparing the latest frame with the past frames or by comparing the input image with the updated map [111].

(b) Stereo Cameras: Stereo and RGB-D cameras measure the distance between objects and camera to overcome the shortcomings of monocular cameras. A stereo camera comprises multiple synchronized monocular cameras [112]. Each pixel's 3D position is calculated from the physical distance from a baseline [68,113]. Stereo cameras require computational power to calculate image depth, stereo matching, and pixel-disparity to generate a real-time depth map. Depth estimation for stereo cameras compares images from multiple left and right cameras. Stereo cameras are used both indoors and outdoors, but are limited by baseline length, camera resolution, and calibration accuracy [102].

(c) RGB-D cameras: These are superior since they can measure distance and build a point cloud from a single image frame. By combining VO and LiDAR sensors, depth information provided by RGB-D cameras can be enhanced [69,102]. RGB-D cameras suffer from:

- Narrow measurement range [102].
- Susceptibility to noisy data [69].
- Small field of view [71].
- Susceptibility to interference [114].
- Inability to detect transparent material.
- Low accuracy in 3D reconstruction and scene understanding in dynamic, unstructured, complex, uncertain and large-scale environments [115].

(d) Fish-eye camera: This is a wide-angle perspective camera used to create a fish-eye view. These cameras cover up to 120°–180° horizontal field-of-view. However, radial lens distortions cause nonlinear pixel mapping, making image processing algorithms more complex [116].

(e) Rolling shutter cameras: These are dynamic vision sensors which produce up to one million frames per second [117]. These integrate camera videos, motion sensors (GPS/IMU), and a 3D semantic map dependent on the environment. Edges, planes, and surface features of the target landmark can be captured by these cameras deploying enhanced feature dependencies and tracking joint edges. In large-scale scenarios

such as in smart cities, rolling shutter cameras that capture geometric features such as points, lines, and planes are used to infer the environmental structures [11].

(f) LiDAR cameras: In SLAM, LiDAR or range-finding sensors generate mapping data based on visual feature matching and multi-view geometry to find overlapping points amidst dense data. To recognize already visited places in landmarks, compact point cloud descriptors are compared between two matching points [118]. LiDAR cameras are used for submap matching as LiDAR scans and point-clouds can be clustered into submaps [119]. LiDAR has sparse, high precision depth data while cameras have dense, but low precision depth data [105]. However, LiDAR does not capture images, and is not well suited to sense see-through surfaces or underwater small objects [120].

In addition to stereo cameras and LiDAR, the use of submaps enables SLAM to account for loop closures and redundant data, as well as minimize the error accumulated by local SLAM to produce accurate point clouds [81]. A point cloud is a vector representation of feature extraction for the post-processing of raw data to produce usable results [121]. The data appears as a collage of different point clouds from different sensors to measure distance, area, and volume of an element in 3D. The disparity map between the images is computed for each radar, LiDAR, and camera sensor to detect moving objects [120]. UAV-based mobile mapping SLAM systems use LiDAR mounted on UAVs to scan confined spaces.

Tracking and combining multiple sensor modalities in mobile mapping systems needs integrating flat cameras, 360° wide angle cameras, IMUs, visual sensors, etc. This is for positional trajectory alignment, estimating UAV position changes and to update positions relative to features [98]. In addition, data from external sensors such as GPS or 3D Li-DAR can also be used to map and navigate using coordinate points [16]. However, as IMU and VO sensors drift over time, multiple fusion instances are required until a desired accuracy is achieved to map and navigate a UAV simultaneously [97,118]. Note also that some applications combine visual and thermal cameras with IMUs in visually degraded environments.

## 4. Simultaneous Localization and Mapping (SLAM)

As mentioned in the previous section, the objective of data fusion is to generate a meaningful set of information $\mathcal{I}$ from the sensor data vector $\mathbf{d}_{n,j}$. This information $\mathcal{I}$ is processed through a set of algorithms known as SLAM to estimate the location of the UAV in a given environment and the environment itself. This is done based on the raw data obtained from the UAV sensors [78]. The SLAM tracks the UAV path and maps landmark locations using a point cloud [70]. The search area on the reference map expands dynamically depending on failed or correct landmark matches [94] as depicted in Figure 3. The symbols and parameters used in the paper are briefly described in Table 2.

**Table 2.** Definition of symbols and parameters used in the paper.

| Symbol | Definition |
| --- | --- |
| $\mathbf{q} = [\mathbf{q}_1, \mathbf{q}_2, \cdots, \mathbf{q}_n]$ | UAV trajectory at different time steps |
| $\mathbf{q}_n = [q_x, q_y, q_z, \theta]_n^{\mathrm{T}}$ | $q_x$, $q_y$, $q_z$ represent UAV position along the three axes and $\theta$ is the angle |
| $\mathbf{q}_n \overset{\Delta}{=} \{q_n, l_1, \ldots, l_m\}$ | Approximate trajectory while capturing landmarks data $l_1, \ldots, l_m$ |
| $q_1 = [q_{10}, q_{11}, \cdots, q_{1n}]$ | Data for UAV 1 from time t = 0 to t = $n$ |
| $\mathcal{L} = [\mathbf{l}_1, \mathbf{l}_2, \cdots, \mathbf{l}_L]$ | Gaussian distributed landmark data (state variables) at time $n$ |
| $\mathcal{I}$ | Set that contains information about UAV trajectory, position, and landmarks |
| $m$ | Number of landmarks in a scene |
| $\mathbf{d}_{n,j}$ | Sensor data |
| $\psi(n) = [\psi_1, \psi_2, \cdots, \psi_n]$ | UAV trajectory; describes how $\mathbf{q}$ changes from time step $n-1$ to $n$ |
| $\psi_1, \ldots, \psi_N$ | Position in the vector space; $n$ dimensional vector for feature detection |

**Table 2.** *Cont.*

| Symbol | Definition |
|---|---|
| $\mathcal{N}_n$ | Noise in sensor data |
| $f(\cdot)$ | Function that describes the SLAM process |
| $P(\mathbf{q}_n \mid \mathbf{q}_1, \mathcal{N}_{1:n}, \mathbf{d}_{1:n})$ | Data from 0 to $n$; estimates the current state distribution at time $n$ |
| $P(\mathbf{d}_n \mid \mathbf{q}_n)$ | Estimated likelihood of sensor data given a UAV trajectory |
| $P(\mathbf{q}_n \mid \mathbf{q}_1, \mathcal{N}_{1:n}, \mathbf{d}_{1:n-1})$ | Prior estimated probabilities of UAV localization |
| $\mathcal{K}$ classes, $K_1, K_2, \cdots, K_n$ | Sequences of UAV trajectory and odometry dataset |
| $\mathcal{T}\mathcal{F}$ | Timeframes at which landmark data is collected/discarded |
| $\mathbb{R}^x \times \mathbb{R}^y \times \mathbb{R}^z$ | Data captured by RGB cameras is pixel information of size $\mathbb{R}^x \times \mathbb{R}^y \times \mathbb{R}^z$ |
| $\tilde{s}$ | Distance between the landmark point and the UAV |
| $\Delta s$ | Distance traversed between coordinates $s_1 = [s_{1x},\, s_{1y},\, s_{1z}]$ to $s_2 = [s_{2x},\, s_{2y},\, s_{2z}]$ |
| $\theta^\circ$ | The orientation difference between two consecutive sensor scans |
| $\Delta\theta^\circ$ | Error in the estimated movement between two frames |
| $\phi^\circ$ | Angle between the landmark point and the UAV |
| $\mathcal{T} = [\Delta s, \Delta\theta]$ | Transformation that learns changes in $s$ and $\theta$ |
| $D_1,\, D_2$ | Sensor data |
| $L_1,\, L_2,\, L_3,\, L_4,\, L_5$ | Five landmarks in a scene |
| $[L_1^{(d_1)},\, L_2^{(d_1)},\, L_3^{(d_1)},\, L_4^{(d_1)},\, L_5^{(d_1)}]$ | Set of sensor data that captures landmark data from the observation space |
| $\left\lVert \mathbf{e}_{14}^{(d_1)} \right\rVert^2,\, \left\lVert \mathbf{e}_{21}^{(d_2)} \right\rVert^2$ | Relative error from sensor data $d_1$ and $d_2$ |
| $\hat{\mathbf{C}}_{n-1}$ | Position covariance |
| $\mathbf{S}_{ij}(\mathbf{q})$ | Sparsity matrix |

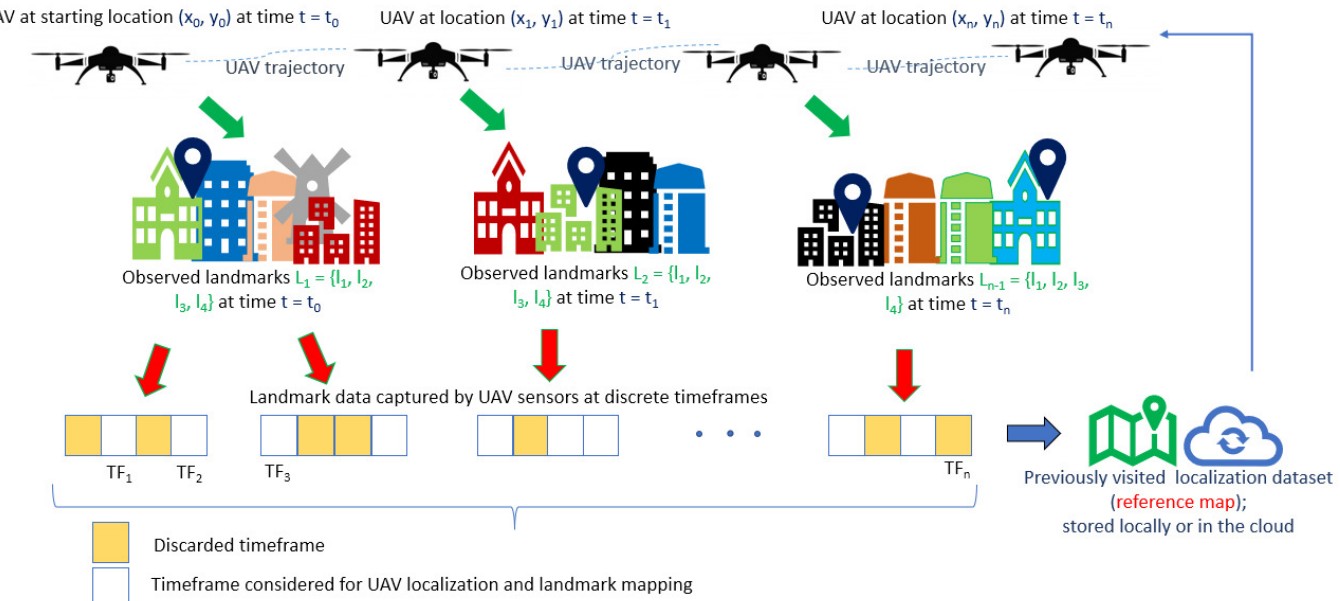

**Figure 3.** From sensor fusion, the most significant and relevant data at each timestep are taken in discrete timeframes for simultaneous localization and mapping (SLAM).

### 4.1. SLAM in UAV: Motivation and Requirements

SLAM begins with raw data collected from various sensors where a UAV, or a network of UAVs, mounted with sensors supposed to move in an unknown environment [2]. A UAV requires path planning, path following, trajectory control, and motion tracking based on odometry readings [47]. Data collection at different times and locations, along with mapping these instants of time, is a continuous process. Data sampling converts this to a set of discrete timesteps $1, 2, \cdots, n$. As shown in Equation (1), the positions based on the trajectory of a UAV are denoted by $\mathbf{q}$. The map comprises several landmarks and at some

timesteps, the sensors can see a part of the landmarks and record their observations [122]. Let there be $\mathcal{L}$ landmarks in the map denoted by $\mathbf{l}_1, \mathbf{l}_2, \cdots, \mathbf{l}_L$ [123].

1. The UAV trajectory describes how $\mathbf{q}$ as $\mathbf{q}_1, \mathbf{q}_2, \cdots, \mathbf{q}_n$ changes from time step $n-1$ to $n$ [124]. The position $\mathbf{q}$ consists of the translation on the three axes and the rotation around the three axes, having six degrees of freedom, and a vector in $\mathbb{R}^6$. The vision sensor variables are usually represented as a vector in $\mathbb{R}^6$ and the $m$ landmarks as a vector in $\mathbb{R}^{3m}$. A photograph of the landmarks in space is projected using the camera's projection model, utilizing state estimation and distortion estimation [74].

2. The sensor observations detect specific landmarks $\mathbf{l}_j$ at trajectory position $\mathbf{q}_n$ [94]. A group of point clouds can fall in the same category if the resolution of the sensor is less than $\phi^\circ$, which is the angle between a landmark point and the UAV [102]. Considering all the sensors of a $360^\circ$ rotation range, the depth values can be stored into a vector, where each possible category is represented by the elements in the vector. Two 1D vectors from sequential sensor scans can be concatenated to provide input for the fusion algorithm. For instance, an image of size $\mathbb{R}^x \times \mathbb{R}^y \times \mathbb{R}^z$ represents an acquisition of the sensor that detects and extracts the features in the surrounding environment [53].

3. SLAM as a state estimation problem estimates the internal, hidden state variables from the sensor data. The UAV trajectory and the sensor observations may be linear or nonlinear and the noise $\mathcal{N}$ in sensor data may be Gaussian or non-Gaussian [73].

$$\mathbf{q}_n = f(\mathbf{q}_{n-1}, [\boldsymbol{\psi_n}], \mathcal{N}_n), k = 1, \cdots, \mathcal{K} \tag{1}$$

where $[\boldsymbol{\psi_n}] = [\psi_1, \psi_2, \cdots, \psi_n]$ is a vector that represents the input sensor data at time $n$, $\mathcal{N}_n$ is noise at time $n$, and the function $f(\cdot)$ describes the SLAM process [96]. The noise $\mathcal{N}_n$ at each timestep is random and is a practical assumption that makes $f(\cdot)$ a stochastic process. If a UAV is instructed to move forward 10 cm, it may move by 9.8 cm or 10.2 cm. As error is accumulated, the UAV's estimate of the position variance increases the uncertainty about its location. For a UAV in space, its position is described by the $x$, $y$, $z$ coordinates and an angle, i.e., $\mathbf{q}_n = [q_x, q_y, q_z, \theta]_n^\mathsf{T}$, where $q_x$, $q_y$, $q_z$ are positions on three axes and $\theta$ is the angle [108]. As shown in Equation (2), the change in position and angle between two timestamps is $\mathbf{\Delta q_n} = [\Delta q_x, \Delta q_y, \Delta q_z, \Delta\theta]_n^\mathsf{T}$. From Equation (2), for total timestamps $n$, $\mathbf{q_n} = [q_x, q_y, q_z, \theta]_n^\mathsf{T}$ is given by:

$$\begin{bmatrix} q_x \\ q_y \\ q_z \\ \theta \end{bmatrix}_n = \begin{bmatrix} q_x \\ q_y \\ q_z \\ \theta \end{bmatrix}_{n-1} + \begin{bmatrix} \Delta q_x \\ \Delta q_y \\ \Delta q_z \\ \Delta\theta \end{bmatrix}_n + \mathcal{N}_n \tag{2}$$

The UAV trajectory can be represented by weighted samples, and a map is computed analytically over a smaller trajectory in the submap, where each landmark is independent [119]. For $m$ landmark locations, $n$ filters compute UAV position distribution. For each of the $m$ landmarks, the captured sensor data $\psi = (\psi_1, \psi_2, \ldots, \psi_m)$ are an $m$ dimensional vector with feature detection and feature matching to estimate the UAV positions over time. The labeled data in already visited places based on scene geometry and reprojection error are used to avoid loop closures [20]. Considering the image sequences in the sensor input in the aerial scenario, each sample is assigned an importance weight $w(i) = P(\mathbf{q}_{(i)n} \mid d_{1:n}, \mathcal{N}_{0:n})$. The samples with low importance weight are replaced by samples with a higher weight [25]. Each sample updates the observed landmarks to determine the spatial relationship [125]. For example, if the UAV is mounted with a sensor that observes a 2D landmark, it measures the distance $\tilde{s}$ between the landmark point and the UAV, and the angle $\phi$. If the landmark is at position $\mathbf{l}_j = [l_1, l_2]_n$, the UAV trajectory is $\mathbf{q}_n = [\mathbf{q_1}, \mathbf{q_2}]_n^\mathsf{T}$,

where $[\mathbf{q_1}] = (q_{x_1}, q_{y_1}, q_{z_1})$, $[\mathbf{q_2}] = (q_{x_2}, q_{y_2}, q_{z_2})$, $\cdots$, $[\mathbf{q_n}] = (q_{x_n}, q_{y_n}, q_{z_n})$. If the observed data is $\mathbf{d}_{k,j} = [d_{k,j}, \phi_{k,j}]^T$, then,

$$\begin{bmatrix} d_{k,j} \\ \phi_{k,j} \end{bmatrix} = \begin{bmatrix} \sqrt{(l_{1,j} - q_{1,n})^2 + (l_{2,j} - q_{2,n})^2} \\ \arctan\left(\frac{l_{2,j} - q_{2,n}}{l_{1,j} - q_{1,n}}\right) \end{bmatrix} + \mathcal{N}_{n,j} \tag{3}$$

The Equation (3) describes how SLAM addresses the problem of estimating the UAV trajectory $\mathbf{q}$ and landmark mapping $\mathbf{l}$. The mapping and localization are done in the presence of noise $\mathcal{N}$ in the sensor data $\mathbf{d}$ [75]. When the UAV sensors capture landmark data $\mathbf{l}_j$ at time $n$ and UAV trajectory position $\mathbf{q}_n$, the landmark pixel information is Gaussian distributed with the distribution function $\mathbf{d}_{n,j}, d(\cdot)$:

$$\mathbf{d}_{n,j} = d(\mathbf{l}_j, \mathbf{q}_n, \mathcal{N}_{n,j}), \quad (n,j) \in \mathcal{I} \tag{4}$$

where, $\mathcal{N}_{n,j}$ is noise and $\mathcal{I}$ is a set that contains the information of the position, UAV trajectory and the landmark data [126,127]. Since there are various sensors, the observed sensor data $\mathbf{d}$ may have different formats. The first-order Markov property assumes that the state at time $n$ is only related to the state at time $n - 1$ and is not related to the earlier states. A sensor having a resolution of $\Delta\theta°$ may result in $\mathcal{K}$ possible classes of point-cloud data [80]. Out of these $\mathcal{K}$ classes, $K_1, K_2, \cdots, K_n$ sequences of UAV odometry dataset assists in estimating the UAV trajectory. During this sequence, the 2D sensor may not detect any obstacle and predict safe trajectory, with increments in the translational error for all subsequences $K_1, K_2, \cdots, K_n$ [119]. The error score is the mean of all sub-sequence errors [47]. With the ground truth for each frame inferred from multiple sensors, the average rotation and translation error is dependent on timeframes $\mathcal{TF}$ where it is harder to estimate the UAV trajectory [26,47]. An approximate variation of $\pm\theta°$ between two timeframes $\mathcal{TF}_1$ and $\mathcal{TF}_2$ in the interval $\pm t$ milli-seconds can deviate a UAV off-trajectory.

In a UAV with a set of 2D sensors, a relatively small point cloud may be sufficient for real-time extraction of sensor data to learn the UAV localization. Adding another sensor such as a monocular or a LiDAR camera increases the accuracy by adding images to localize the UAV where many objects are detected [128]. The sensors learn the geometric features, the appearance, and visual context of the landmark scenes in unknown environments. Each observation is a 360° set of points measured during one sensor rotation, and the SLAM algorithm needs to predict the transformation $\mathcal{T} = [\Delta s, \Delta\theta]$, where $\Delta s$ represents the UAV distance traversed when the UAV coordinates change from $s_1 = [s_{1x}, s_{1y}, s_{1z}]$ to $s_2 = [s_{2x}, s_{2y}, s_{2z}]$ and $\Delta\theta$ is the orientation difference between two consecutive sensor scans for UAV states [75]. The displacement of the UAV relies on the mapping of states to $\mathcal{T} = [\Delta s, \Delta\theta]$ at time $n$ and the position $q_x, q_y, \theta_n$ of the UAV [74,129]. With a 2D camera image, the incremental change in UAV position along the axes is $q_x = q_{x_{n-1}} + \Delta s \sin(\theta_{n-1})$, $q_y = q_{y_{n-1}} + \Delta s \cos(\theta_{n-1})$. The Equation (5) represents that the local coordinates of the UAV and the landmark are accumulated to estimate the global position of the UAV at time $n$. The correlation between the ground truth and the estimated translation and rotation errors varies the position and landmark variables. If $\mathbf{q}_n$ is all the unknowns at time $n$ that contains the current UAV position and $j$ landmarks, then:

$$\mathbf{q}_n \overset{\Delta}{=} \{q_n, l_1, \ldots, l_m\}. \tag{5}$$

At time $n$, the data from 0 to $n$ estimate the current state distribution [75]:

$$P(\mathbf{q}_n \mid \mathbf{q}_1, \mathcal{N}_{1:n}, \mathbf{d}_{1:n}). \tag{6}$$

The previously collected data is the reference data for UAV localization and mapping. The subscript $1 : n$ represents all the data from time 0 to time $n$, $\mathbf{d}_n$ represents the sensor

data at time $n$ [83,84]. $\mathbf{q}_n$ is related to the previous states $\mathbf{q}_{n-1}$, $\mathbf{q}_{n-2}$. For state estimation, Bayes' rule implies [72]:

$$P(\mathbf{q}_n \mid \mathbf{q}_1, \mathcal{N}_{1:\,n}, \mathbf{d}_{1:\,n}) \propto \underbrace{P(\mathbf{d}_n \mid \mathbf{q}_n)}_{\substack{\text{Estimated} \\ \text{likelihood}}} \underbrace{P(\mathbf{q}_n \mid \mathbf{q}_1, \mathcal{N}_{1:\,n}, \mathbf{d}_{1:\,n-1})}_{\substack{\text{Prior estimated} \\ \text{probabilities}}}. \tag{7}$$

The current state $\mathbf{q}_n$ is estimated based on the past states. The effect of $\mathbf{q}_{n-1}$ is expanded according to the conditional probability of $\mathbf{q}_{n-1}$ moment [96]:

$$P(\mathbf{q}_1, \mathcal{N}_{1:\,n}, \mathbf{d}_{1:\,n-1}) = \int P(\mathbf{q}_n \mid \mathbf{q}_{n-1}, \mathbf{q}_0, \mathcal{N}_{1:\,n}, \mathbf{d}_{1:\,n-1}) P(\mathbf{q}_{n-1} \mid \mathbf{q}_0, \mathcal{N}_{1:\,n}, \mathbf{d}_{1:\,n-1}) \mathrm{d}\mathbf{q}_{n-1}. \tag{8}$$

### 4.1.1. Single-UAV SLAM

The state variables of the landmarks $\mathcal{L} = [l_1, l_2, \cdots, l_n]$ of a map are acquired by the moving UAV [130]. When a priori map and localization information are not available, positioning systems use SLAM to explore unknown environments. SLAM maximizes the posterior of the mapping data and the corresponding UAV state [131] by either:

- filtering: updating the current state at each time step given the new observation (Using the past and future information to update the current state is called batch filtering) [95].
- smoothing: optimizing the whole trajectory based on the past observations (Using only the past information to update the current state is called incremental smoothing) [95].

SLAM for a single UAV finds the solution of the maximum a posteriori (MAP) based on prior distribution obtained by UAV odometry over the trajectory $\mathbf{q}$. The posterior distribution obtained with Bayes' theorem as given in Equations (7) and (8) is the SLAM likelihood given a certain prior distribution of the UAV motion state [75]. Enough memory is required to maintain the trajectory of the UAV in a given state [59]. The updated UAV states indicate where the UAV needs to be at that moment in order to arrive at its destination precisely in the shortest time [60].

### 4.1.2. Collaborative or Multiple UAV SLAM

Many applications need multiple UAVs [33], as depicted in Figure 4. Here, the complexity of the task is shared by multiple UAVs in a given space. To estimate each UAV's position, collaborative SLAM (C-SLAM) and multimodal sensor fusion assist in planning navigation trajectory and mapping an environment by combining the data collected by each individual UAV [5]. C-SLAM, also known as multi-UAV SLAM, can be used for collaborative perception of the UAV environment to enable autonomous control and decision making in unfamiliar and GPS-restricted environments. C-SLAM can be applied to UAV systems by building a collective representation of the environment and sharing situational awareness [56]. The network of UAVs can reduce the necessity for centralized computation and large servers [132]. Each UAV benefits from other UAVs that leads to accurate localization and mapping [57]. UAVs share data to detect if other UAVs have visited the same area, and then estimate an alignment and overlap on the map [94,128].

One example of C-SLAM under communication constraints is evaluated in the DARPA subterranean challenge where no prior information is available about the operating environment [133]. In DARPA mapping applications, C-SLAM is envisioned as a useful tool for two UAVs. UAV state estimation is combined with its surrounding environment in which a moving UAV collects data simultaneously through embedded sensors. The sensor calibration determines the accuracy of the UAV orientation and position [46]. The environment map includes landmark coordinates and orientation, and its accuracy is determined by sensor calibration requiring embedded sensors to continuously collect data [72]. The full mapping data sent to every UAV leads to redundancy, and a subset of UAVs can be designated for computation. C-SLAM solves the estimation problem on each UAV [56]. Each UAV computes its own local map and uses partial information from other UAVs as well as inter-UAV measurements to achieve a local solution. Over several iterations, the

UAV's local solution converges to a solution consistent with the global reference frame [74]. These techniques mitigate communication and computation bottlenecks. All UAVs could send their sensor data directly to a single unit for feature extraction and communicate for data association and state estimation to find links and relative measurements between the individual maps [54,55]. If from time $t = 0$ to $t = n$, the UAV traverses from $\mathbf{q}_0$ to $\mathbf{q}_n$ and collects landmark information $\mathbf{l}_1, \cdots, \mathbf{l}_n$ as for UAV 1, $q_1 = [q_{10}, q_{11}, \cdots, q_{1n}]$ and for UAV 2, $q_2 = [q_{20}, q_{21}, \cdots, q_{2n}]$, then the individual and shared landmark estimates and odometry measurements are combined when a UAV transitions from one time step to the next.

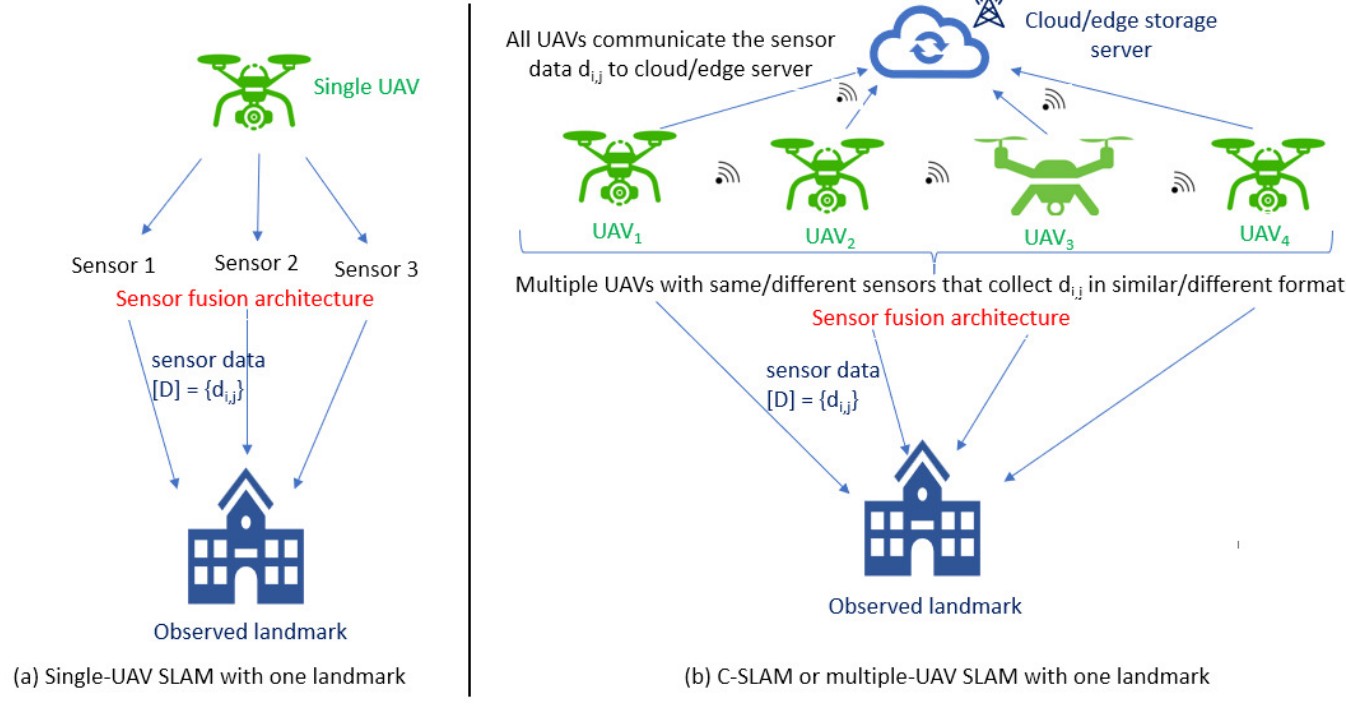

**Figure 4.** Comparative depiction of single-UAV SLAM and C-SLAM with one landmark.

Some advantages of C-SLAM over single-UAV SLAM are as follows:

- Multiple instances of single-UAV SLAM position lead to challenges related to spatial configuration over time dependency on the number of UAVs [134].
- Robust to the loss of individual units [2].
- C-SLAM is more efficient as the number of UAVs increases coordination, communication range and spatial distribution [132].
- The bandwidth available in 5G networks allows information to be continuously and seamlessly transmitted by the UAVs as they move and their trajectory changes [61].
- Distribution of tasks reduces the overall computational cost of C-SLAM [58].

One positive aspect is that the compact representations with high-level semantic features communicate only the object labels and positions to other UAVs [115]. In full connectivity or multiple-hop connectivity, each UAV can directly communicate with other UAVs at any time, or only share information with one UAV at a time [62]. When all the UAVs' initial states can be estimated, the C-SLAM problem is an extension of the single-UAV problem and includes UAVs' states as well as inter-UAV measurements that link multiple UAV maps [75]. As a result, each UAV's neighbors need multiple neighbor-to-neighbor transmissions to reach all UAVs [5,61]. When two UAVs meet, the two maps are transformed according to their relative positions [94]. In non-consecutive timeframes, the UAV trajectory compares positions of several UAVs to draw connections between their local maps and their current relative locations [5]. The disparity map between the

images is computed for obstacle detection. This is done by accumulating the pixels with the same disparity to provide a representation of the geometric structure, which will enable estimation of the ground plane pixels [94]. Classifying the landmarks through pixel positions, by separating the pixels that are over a ground threshold height, projects the pixel values in the image rows and columns [69]. Geometrically matching consecutive images re-localizes and reduces estimation error caused by odometry drift [135]. Semantic information varies between image sequences taken during day and night, or in different seasons. SLAM needs high data rates up to 1 Tbps and a latency less than 1 msec. Massive computation powers at the device level are required in UAVs to achieve sub-centimeter level accuracy of positioning and to construct 3D maps without prior knowledge [107].

### 4.1.3. Limitations of C-SLAM

The relative position estimate of UAV trajectories depends on the starting locations and orientations of each UAV. For alignment among multiple UAV trajectories, a prior distribution formulates C-SLAM as a maximum likelihood estimation (MLE) problem [56]. If each UAV performs some computation, a central node is required to merge the individual data. C-SLAM aims to achieve global perspective from the local perspectives of single-UAV SLAM from sparse or dense maps [134]. The precise topological localization that makes maps interpretable and actionable depends on the number of UAVs involved, the networking limitations, etc. [94]. When the images are not in chronological order, or there are completely unrelated images, the data at the next timestep add noise and uncertainty to the previous timestep [46,99].

C-SLAM considers the global perspective for the position and map of each UAV to estimate the shared global reference frame [50]. When a UAV can partially see a landmark in one location, then $\mathbf{q}_n$ is not known for all $\mathbf{l}_j$. However, each UAV may estimate a set of landmarks using different coordinate systems. The shared positions of observed landmarks need to be in a consistent data format within neighboring UAVs to gradually improve the estimates from the neighbors' latest data [52]. If the local reference frames are in different coordinate systems, then estimating the global reference frame using C-SLAM to collectively perceive the environment needs further pre-processing [62].

UAV localization through sensor image registration maps the information, and conflict arises when two images are alike. To differentiate the states, it is important to identify missing information or conflicting information [74]. Environment mapping using sensors is a map representation that periodically updates the state. Computing the posterior probability of the map based on sensor measurements and deciding how and where to store sensor data is critical for C-SLAM [84]. Prediction calculates the prior probability distribution based on the posterior probability computed one timestep before the state transition [75]. When the UAV returns to a previously explored location known as the loop-closure [136], data association recovers the relative spatial transformation between two timeframes $\mathcal{TF}_1$ and $\mathcal{TF}_2$ [74]. Frame-to-frame motion provides a local estimate of the UAV trajectory, but gradual accumulation of error over time causes inconsistencies. Emphasizing the depiction in Figure 4, Table 3 compares the features and requirements of single-UAV SLAM and C-SLAM. The expectations and challenges in single-UAV SLAM and C-SLAM for UAV applications explore how single-UAV SLAM or C-SLAM can be used to capture the stationary or moving objects in 2D or 3D.

**Table 3.** Expectations and challenges in single-UAV SLAM and C-SLAM for UAV applications [56–58,131].

| Single-UAV SLAM | Multiple-UAV SLAM or C-SLAM |
|---|---|
| Usually one UAV is in action in a given scenario, making it suitable for customized applications. | The number of active UAVs in an application may change dynamically based on resource demands as ad-hoc topology can be created and deployed. |
| The data captured by all the sensors mounted on a single UAV may be critical for fusion. | Targeted data from selected sensors form specific UAVs may be sufficient for fusion. |
| As shown in Figure 2, only one UAV is assigned to all the landmarks in a scene. | Varying number of UAVs can be assigned to each landmark. |
| The available bandwidth is usually fixed and limited. This becomes a critical limitation in single UAV SLAM applications. | As shown in Figure 4, efficient allocation and use of available bandwidth is possible. |
| One possible advantage in some applications is that the uncertainty factor induced by multiple UAVs is alleviated to some extent. However, uncertainty in sensor data still persists. | C-SLAM can lead to precise sensor data from multiple UAVs, but uncertainty may be induced due to multiple UAVs. |
| As depicted in Figure 3, the amount of data gathered by a sensor is limited by the trajectory of one UAV. | C-SLAM allows diverse data collection through multiple UAVs for map construction over the target landmarks. |
| Single-UAV SLAM does not need load balancing for resource provisioning among UAVs. Single-UAV SLAM also needs less computational capabilities for data-offloading across edge/cloud platforms. | C-SLAM requires automatic load balancing and dynamic topology reconfiguration. As shown in Figure 4, C-SLAM also requires additional networking functionalities for collaboration among multiple UAVs and cloud infrastructure. |
| Due to limited FoV, the sensors may intermittently not be able to capture data along one or more dimensions unless the orientation of a single UAV allows sufficient FoV. Consequently, a target landmark may be viewed from only one angle at a time. | The missed data by one UAV may be compensated over a series of timesteps by other UAVs for a large number of landmarks. Furthermore, a target landmark can be viewed from multiple angles. |
| Due to limited battery power, single-UAV SLAM may not be feasible in large trajectory scenarios. | C-SLAM may benefit from dynamic UAV placement based on least-congested or shortest trajectory for each UAV, thus optimally utilizing available battery power. |
| The operational costs may be reduced due to less equipment. | More operational costs due to infrastructure requirements. |
| Manual control and restoration may be needed in case of UAV failure. | Automated and flexible restoration techniques can be feasible in case one or more UAVs fail. |

*4.2. Search Space Reduction in Linear Systems Using Kalman Filter*

When the Markov property is assumed, the current state is only related to the previous moment. The estimation of the posterior at time $n$ only depends on the posterior at time $n-1$ [137]. Assuming Markov property, let the posterior state estimation at time $n-1$ be denoted by $\hat{\mathbf{q}}_{n-1}$ and its covariance $\hat{\mathbf{C}}_{n-1}$. To determine the posterior distribution of $\mathbf{q}_n$ based on the observations at time $n$ needs to distinguish the prior and the posterior variables [63]. In featured-based methods, an input image is abstracted to a group of features sequentially tracked to estimate the state of the UAV. The image features represent

landmarks in the map [73]. As represented in Equations (9) and (10), the Kalman filter updates the full state vector with the UAV position and the feature locations, assuming the states earlier than $n - 1$ are not related to the $n$th state [138]. The linear system can be solved using Kalman filters (KF) and the nonlinear non-Gaussian system can be solved using extended Kalman filters (EKF), where the discontinuities in scanning landmarks are filtered and corrected [139].

$$P(\mathbf{q}_n \mid \mathbf{q}_{n-1}, \mathbf{q}_0, \mathcal{N}_{1: n}, \mathbf{d}_{1: n-1}) = P(\mathbf{q}_n \mid \mathbf{q}_{n-1}, \mathcal{N}_n) \tag{9}$$

$$P(\mathbf{q}_{n-1} \mid \mathbf{q}_0, \mathcal{N}_{1: n}, \mathbf{d}_{1: n-1}) = P(\mathbf{q}_{n-1} \mid \mathbf{q}_0, \mathcal{N}_{1: n-1}, \mathbf{d}_{1: n-1}) \tag{10}$$

For $\mathcal{N}_n$ to be independent of $\mathbf{q}_{n-1}$, the state distribution at time $n - 1$ is used to derive the state distribution from time $n - 1$ to time $n$, to estimate and update the current state incrementally [140]. Under the assumption that the states and noise are Gaussian, for Kalman filters, the estimated pose, mean and uncertainty in covariance can be represented as [73]:

$$\begin{cases} \mathbf{q}_n = \mathbf{A}_n \mathbf{q}_{n-1} + \mathbf{q}_n + \mathcal{N}_n \\ \mathbf{d}_n = \mathbf{B}_n \mathbf{q}_n + \mathcal{N}_n \end{cases} \quad n = 1, \ldots, N, \tag{11}$$

where $A$ and $B$ are constants that make a matrix Hessian and sparse. The sparsity property optimizes the computational complexity required for maximum posterior probability estimation and unbiased state estimation [141].

### 4.3. Search Space Reduction in Nonlinear Systems Using Extended Kalman Filters

A filtering technique widely used to solve nonlinear tracking problems is the extended Kalman filter (EKF). EKF are Gaussian filters that are based on local linear approximation of Kalman filters [73]. This linearization may lead to inconsistencies under noise. An advantage of EKF is that the covariance matrix does not require additional computations for feature tracking or active trajectory exploration in uncertainty [95]. A covariance matrix is required to assume filters with smooth variables. Linearization error that arises from the marginalization of past variables requires new links among the remaining variables. Note also that the elimination of state variables leads to reduced interdependence between landmark variables [51,142].

Increasingly coupled variables require more computation with less marginalization when the landmark variables are sparsely connected [60,79]. Smoothing techniques increase SLAM accuracy when a UAV revisits the past locations that are added to the latest estimate. Sparsity reduces the amount of data exchange during estimation to continuously update global SLAM [143]. Gaussian elimination requires an exchange of dense marginals, and the computational complexity varies in a quadratic manner based on the number of inter-UAV measurements for convergence in noiseless scenarios. Sparse semi-definite relaxation provides exactness in moderate noise. Meanwhile, the majorization-minimization technique enhances the accuracy of SLAM estimation in the presence of outliers; known as perceptual aliasing when two different places are inferred as the same [144]. In EKF based SLAM, the required number of computations also increases in a quadratic manner with the number of landmarks [66]. Outlier mitigation leads to mutually consistent expectation maximization of inter-UAV measurements. Pairwise consistency maximization eliminates loop closures [136] by detecting overlaps where the viewpoint and lighting conditions are similar, as they depict the same place viewed by the UAV at the same time. The UAV motion and landmark observations in SLAM are usually nonlinear functions [145]. A UAV capturing images is approximated by non-linear models and the sensor measurements are also nonlinear.

The steps involved in the search space reduction in nonlinear systems using extended Kalman filters are outlined as follows:

- The UAV position $\hat{\mathbf{q}}_{n-1}$ and the position covariance $\hat{\mathbf{C}}_{n-1}$ are predicted first [140].
- The EKF method assumes Markov property, i.e., the $n$th state of a UAV is related to observations and state at time $n$ and not on time $n - 1$. For two adjacent frames in

VO, if the current frame is related to data before $n - 1$, then the EKF requires more computations [51,146].

- A large point-cloud depends on the non-linearity of the observations and the linear approximation is valid in a small range [70,80]. For multiple submaps [119], the EKF filter is linearized at every $\hat{\mathbf{q}}_{n-1}$ that leads to nonlinear error in EKF [147]. Estimation of spatial uncertainty in the UAV trajectory uses EKF to estimate the mean and covariance of the states [74].
- The EKF stores the state variable's mean and variance in the memory. If the number of landmarks is significantly larger than the UAV trajectory positions [148], then the storage grows in a quadratic manner with the number of states. As the covariance matrix is also stored, EKF SLAM is not suitable for large-scale scenarios to estimate $\hat{\mathbf{q}}_n$ from previous $n - 1$ observations. The complexity increases further when it is required to simultaneously update the state vector of UAV maps based on the covariance $\hat{\mathbf{C}}_{n-1}$ of landmark $m$ [146].
- Incorrect observations added to EKF cause it to diverge [146]. EKF lacks an outlier detection mechanism, which causes the system to diverge when there are outliers. Outliers in visual SLAM can match a UAV to the wrong target. Lack of an outlier detection mechanism makes the system unstable [66].

As depicted in Figure 5, a question arises that a UAV can return to its starting position in GPS-enabled or GPS-denied environments? If a UAV cannot use GPS, it may follow previous paths or take a new route to return to its starting position. Will it pass through previous locations or will it take a new path, and how will outliers be dealt with [22,23,47]? In recursive estimations, drift occurs when errors are accumulated over time. The error in prior estimations has a significant impact on the new estimations [92]. EKF is effective when computing resources are limited or the state dimension is relatively small [139]. Another variation known as the unscented Kalman filter (UKF) aims to address challenges related to consistency, filter convergence, and data association in SLAM [66]. The UKF avoids linearization through mean and covariance parameterization [95] and assigns observations to landmarks using maximum likelihood. The UKF does not represent the uncertainty and effect of an observation from data association. Batch gating reduces the effect of wrong data association by exploiting the geometric relationship between landmarks. Two landmarks that are far from each other are weakly correlated by dividing the map into smaller submaps and assigning a smaller UKF to each submap [119].

As a proposed solution in robotics and autonomous vehicles, particle filters complement the SLAM-based navigation systems with absolute position estimation. The particle filtering process updates the states during each window around each particle during a measurement timeframe $\mathcal{TF}$ [138]. To address the shortcomings of EKF, such as the linearization error and noise due to Gaussian distribution assumptions, particle filters have been used while a UAV navigation is mapped using the estimates provided by the sensors [149]. The inter-UAV measurements are based on Euclidean distance between the particle descriptors and the current landmark image descriptors [150]. In particle filtering, the mean $\boldsymbol{\mu}$ and covariance $\mathbf{C}$ are computed from the distribution of the particles in each window [140]. A particle filter, also known as a sequential Monte-Carlo (SMC) filter, represents the probability distribution as a set of samples where each sample approximates the true value of a state. In an interesting example [140], Rao-Blackwellized particle filters (RBPF) use samples to represent the posterior distribution, and to perform variable marginalization using an EKF to reduce the size of the sampling space. The Table 4 compares the features and requirements of Kalman filters, extended Kalman filters, unscented Kalman filters and particle filters for UAV-SLAM. A brief comparison of key features of various filters for state estimation in UAV-SLAM reveals that it is possible to calculate ground entity position relative to UAV trajectory based on these filters, known UAV trajectory orientations and distance from ground-based landmarks.

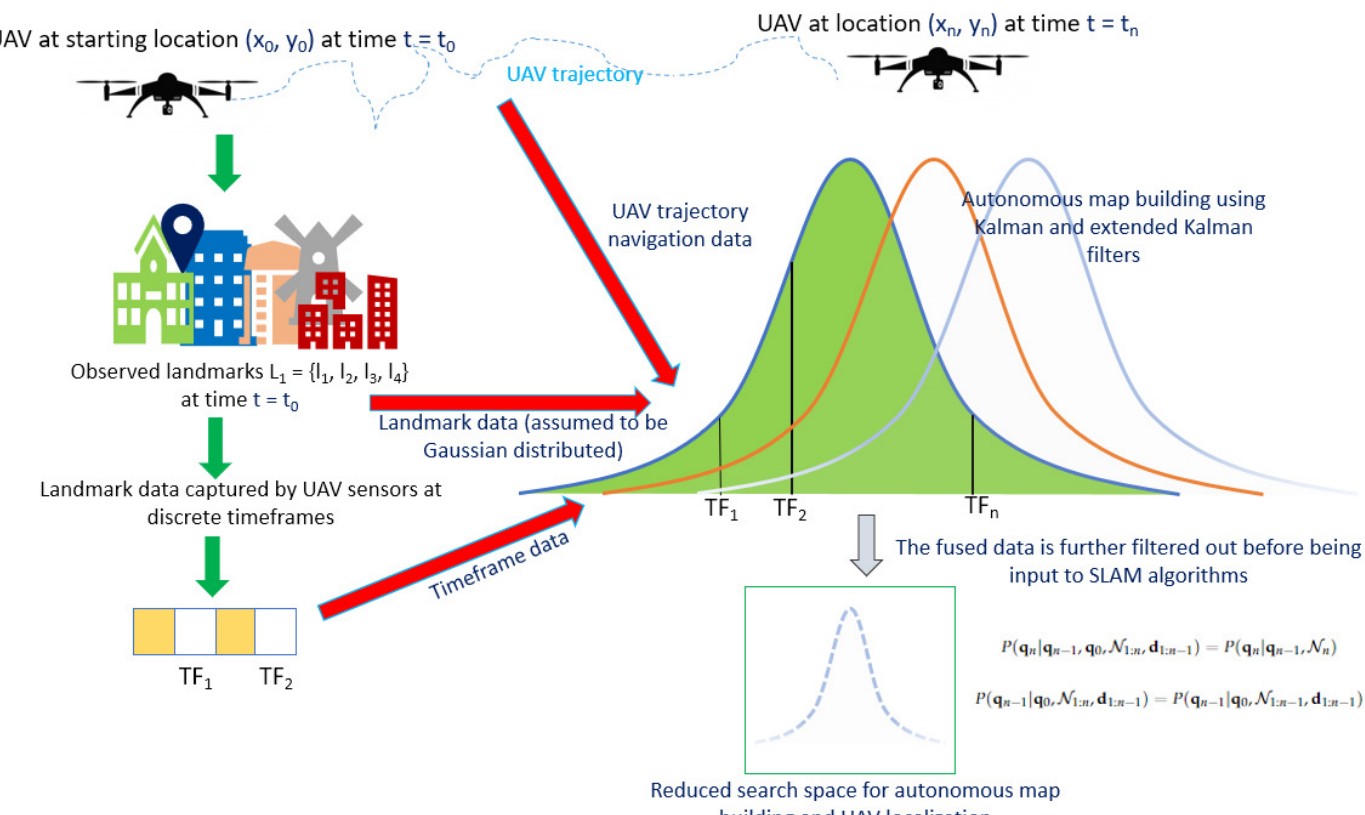

**Figure 5.** Kalman and extended Kalman filter based search space reduction and localization of UAVs.

**Table 4.** A brief comparison of key features of various filters for state estimation in UAV-SLAM [73,95,140].

| Kalman Filters | Extended Kalman Filters | Unscented Kalman Filters | RBPF |
|---|---|---|---|
| In simple scenarios, if the UAV trajectory is linear, it is relatively easier to analytically obtain UAV state-estimation. | As depicted in Figure 5, EKF is applicable for non-linear UAV trajectory and state-estimation. | UKF simplifies non-linear UAV trajectory using statistical linearization and may require fewer computations compared to EKF. | These filters address non-linear UAV trajectory recursively through Monte-Carlo statistical state estimation. |
| Kalman filters are computationally cheaper to implement in Matlab or python using matrix algebra. | EKF may be implemented as a Kalman filter that applies piece-wise linearization of non-linear sensor data. | UKF can further reduce the required computational power. These can be effective for C-SLAM. | Can be used in UAV-SLAM for non-Gaussian tracking of UAV trajectory and landmark data. |
| Generally provides optimal solution when data may be non-Gaussian and noise is Gaussian distributed. | Requires sensor data and noise to be Gaussian distributed. | UKF does not require noise and sensor data to be Gaussian distributed. | Suitable for multimodal sensor fusion where data are nonlinear and non-Gaussian, and the posteriors are randomly distributed. |

## 5. Visual SLAM and Image Fusion

In visual SLAM, the features are extracted using images. Visual SLAM localizes and builds a map using these images that continuously outputs UAV positions and landmark maps [4,149]. Visual UAV odometry and navigation trajectory information is used as input for sampling steps in visual SLAM. Initialization, tracking and mapping define a coordinate system for UAV position estimation and 3D landmark reconstruction to

continuously estimate 2D-3D correspondence between image and map obtained from feature matching [148]. In visual SLAM, the sensor is a RGB-D camera that captures the pixels in the landmark images [69]. The cameras capture a small part of the landmarks at a given time [114]. The information about a landmark location and how other locations are connected is used to create a feature-based map that differentiates the static and dynamic landmarks. Visual SLAM helps to detect landmarks and add them to a map with the location information that is sequentially updated with new observations [64]. For a large number of feature points, assuming the sensors move at a constant speed, estimating the landmark distances through a set of images need not be accomplished in a chronological order. Automating point cloud classification to identify objects in a scene by integrating additional sensors, visual SLAM complements thermal imagery or full spectrum imagery to capture diverse landmark data [84]. A visual SLAM functionality in UAV includes the following steps [151]:

1. Sensor data fusion: Acquisition, processing, and synchronization of sensor data and images [84].
2. Visual odometry: Estimates the sensor movement between adjacent frames to generate a UAV trajectory map [99].
3. UAV trajectory optimization: The UAV position and landmark data at different timesteps generate an optimized trajectory and map for real-time SLAM [4,94].
4. Place recognition and loop closing: Place recognition determines if a UAV has reached a previously visited position and loop closure reduces the accumulated drift [135].
5. Map reconstruction: Constructs an application-specific map based on the estimated UAV trajectory [5,136].

*5.1. Visual Odometry and Photogrammetry in UAV*

Visual odometry is concerned with the movement of a camera between adjacent image frames as a sensor is rotated or translated by $\theta°$ or $x$ centimeters [152]. However, the error can increase to the order of a few kilometers in smart city scenarios, where tall and small buildings are clustered together. Furthermore, the error is shown to increase exponentially with time. Pixels are the projections by spatial points onto the camera's imaging plane [102]. Each pixel's 3D location is determined by the camera's position at each time step [80,128]. This helps to estimate the UAV motion from images of adjacent frames, to re-create the 3D scenes, and to estimate sensor movements between two successive frames. By calculating the UAV motion from the adjacent frames and restoring the 3D scene, the sensor movements between successive frames can increase the precision of the estimated SLAM. The approximate 3D position of each pixel and the UAV trajectory can be used to update the map. Visual SLAM uses data from cameras for real-time photogrammetry and VO. To enhance visual SLAM performance, data fusion using GPS, Wi-Fi, multi-angulation, sonar, and radar processes raw data via feature recognition, which reinforces the identified landmarks and objects [23]. Visual odometry provides the landmark change between two timeframes [27]. Due to error, an estimated change in $\theta°$ may have an error of $\Delta\theta°$ in real-world applications. Even if the following estimates are correct, the $\Delta\theta°$ error in the trajectory will be accumulated in relative motion estimation [122].

Visual SLAM can be interpreted as perceiving a UAV environment to create a precise localization by modeling the sensor–fusion characteristics based on captured scene geometry. In urban environments, a large amount of information needs to be processed in real-time amidst multiple dynamic obstacles [123]. Feature-based maps identify and extract landmark features to build a map that provides a compact representation of a structured environment [153]. Metric maps emphasize the exact locations of the landmarks in maps classified as sparse or dense. Metric maps represent the geometric properties of the environment, such as the distance between UAVs, or the distance between a UAV and a building. Sparse metric maps store the landmark scene in a compact form, do not express all the objects, and select a few representative landmarks while maintaining the photo-metric consistency and geometric consistency of feature positions [124]. Dense metric maps focus

on modeling all the landmarks. In UAV SLAM, a sparse map can be used for localization, while a dense map is preferred for navigation. Occupancy grids emphasize the relationships among map elements composed of nodes and edges, considering the connectivity between landmark objects [124]. SLAM requires precise UAV location while simultaneously mapping an environment, yielding a compact map by removing extraneous details.

In C-SLAM, maps are used to find partial overlaps for places that have been visited by multiple UAVs. A UAV trajectory can be used as a reference coordinate system in urban mapping. Visual SLAM based mobile mapping systems in UAVs capture multiple image datasets at the same time, integrating 360° imagery [15,17]. Visual SLAM finds a set of state variables, environment landmarks and UAV trajectory from sensor measurements to update a previous estimation [75]. To enrich the UAV's knowledge of an environment, various landmark features, such as the upper part of a building, statues, mechanical rooftop apparatus, etc., are recorded with the photogrammetry [154]. UAV photogrammetry data enhances interactivity and information sharing. Images captured from terrestrial and aerial photogrammetry for feature extraction enable data association for the joint estimation of the map and the UAV state [155]. Figure 5 illustrates the computation of the UAV position at time $n$. The distribution shows the joint posterior density of the UAV position $q$ at any time $n$, and the landmark positions $l$, based on the observations and motion from time 0 to $n$ with respect to the initial position $q_0$ [142]. To fix misalignment issues that may come up with the imagery data, the point clouds may be colorized to enhance the level of detail and accuracy of image sensors [156]. In order to create a point cloud with static scanning accuracy $\pm x$ millimeters, a UAV needs to reset the sensor scanner and post-process the data into an exportable point cloud [125]. For example, measuring the width of a building or the circumference of a light pole requires the resetting of the scanner every few hours of scanning before exporting the data into a usable point cloud. Current UAV mobile mapping systems execute one continuous capture and export the data in a universal format that can be interpreted by a third-party software [156]. Table 5 summarizes the discussion on visual SLAM and aerial photogrammetry in UAV.

**Table 5.** Comparison of generic SLAM and visual SLAM in UAV [148].

| SLAM Architecture Based on All Types of Sensor Data | Visual SLAM and Aerial Photogrammetry |
| --- | --- |
| UAV trajectory and the landmark map are captured through various sensors. | UAV trajectory, localization and landmark map-building rely primarily on visual sensing. |
| Different sensors can capture data of varying size to capture the landscape that changes drastically between two timeframes. | Visual SLAM primarily generates pixels or point clouds to depict randomly varying landmark geometry. |
| The range up to which the sensors can sense landmarks may be limited, and the measurements may be impacted by noise. | Increased range offered by LiDAR cameras to sense a large number of landmarks that vary in size and distance from UAV. |
| The relationships between UAV trajectory and landmark positions can be stored in 1D or 2D. | Visual SLAM and aerial photogrammetry can capture occluded landmarks through point cloud matching. |
| The resulting maps usually contain coordinate information about landmark locations. | The resulting maps contain visual information as well as coordinate information about landmark locations. |

*5.2. Impact of Sensor Parameters on Accuracy of Visual 3D Reconstruction*

Visual 3D reconstruction needs optimizing sensor parameters as well as 3D landmark images [44]. The light rays emitted from 3D points projected into the image planes of several image sensors and cameras are detected as feature points [150]. UAV positioning and map refinement need parallel motion estimation and mapping SLAM at various

timeframes [157]. Visual SLAM is based on the sparse structure of the error term $\mathbf{e}_{ij}$ that describes the residual about $\mathbf{l}_j$ in $\mathbf{q}_i$, the $i$-th position and the $j$-th landmark [148]. The derivatives of the remaining variables are 0. The corresponding error term in the sparsity matrix containing the visual SLAM imagery data has the following form [25,64,125,141,149]:

$$\mathbf{S}_{ij}(\mathbf{x}) = \left( \mathbf{1}_{3\times6}, \ldots \mathbf{0}_{3\times6}, \frac{\partial \mathbf{e}_{ij}}{\partial \mathbf{q}_i}, \mathbf{0}_{3\times6}, \ldots \mathbf{0}_{3\times6}, \ldots \mathbf{0}_{3\times6}, \frac{\partial \mathbf{e}_{ij}}{\partial \mathbf{l}_j}, \mathbf{0}_{3\times6}, \ldots \mathbf{0}_{3\times6} \right), \quad (12)$$

where $\mathbf{1}_{3\times5}$ represents the **1** matrix with dimension $3 \times 6$; and similarly $\mathbf{0}_{3\times5}$ represents the **0** matrix with dimension $3 \times 6$. The dimension 3 indicates three colors, and the dimension 6 indicates the 6-DoF of image sensors and cameras [46,144]. The partial derivative of the error term in the sensor data related to landmark localization is $\partial \mathbf{e}_{ij}/\partial \mathbf{q}_i$ and has a dimension of $3 \times 6$, and the partial derivative of the landmark $\partial \mathbf{e}_{ij}/\partial \mathbf{l}_j$ dimension is also $3 \times 6$ [143]. The error term is independent of other landmarks and UAV positions except at $\mathbf{q}_i$ and $\mathbf{l}_j$. If this observation edge is depicted only as two vertices, then the sparsity matrix $\mathbf{S}_{ij}$ has non-zero blocks in the column $i$, $j$. The corresponding non-zero blocks are at $(i, i)$, $(i, j)$, $(j, i)$, $(j, j)$:

$$\mathbf{S} = \begin{bmatrix} \mathbf{S}_{11} & \mathbf{S}_{12} \\ \mathbf{S}_{21} & \mathbf{S}_{22} \end{bmatrix}. \quad (13)$$

where $\mathbf{S}_{11}$ is only related to the UAV trajectory and $\mathbf{S}_{22}$ relates only to landmark data. $\mathbf{S}_{12}$ and $\mathbf{S}_{21}$ may be sparse or dense, depending on the specific observation data. The **S** matrix computation takes advantage of the sparsity [158]. Suppose there are two sensor datasets $(D_1, D_2)$ and five landmarks $(L_1, L_2, L_3, L_4, L_5)$ in a scene. $D1$ is a set of sensor data from the observation space $[L_1^{(d_1)}, L_2^{(d_1)}, L_3^{(d_1)}, L_4^{(d_1)}, L_5^{(d_1)}]$ and $D2$ is the set of sensor data from the observation space $[L_1^{(d_2)}, L_2^{(d_2)}, L_3^{(d_2)}, L_4^{(d_2)}, L_5^{(d_2)}]$. The variables corresponding to these sensors and point clouds are $\mathbf{d}_i, i = 1, 2$ and $\mathbf{l}_j, j = 1, \cdots, 5$ [70,82]. The sensor that captures data $D_1$ observes the landmarks $L_1, L_2, L_3, L_4$, and the sensor that captures data $D_2$ observes $L_1, L_3, L_4, L_5$. If the $i^{th}$ sensor can observe the $j$-th landmark, the overall cost function is:

$$\frac{1}{M} \left( \left\| \mathbf{e}_{11}^{(d_1)} \right\|^2 + \left\| \mathbf{e}_{12}^{(d_1)} \right\|^2 + \left\| \mathbf{e}_{13}^{(d_1)} \right\|^2 + \left\| \mathbf{e}_{14}^{(d_1)} \right\|^2 + \left\| \mathbf{e}_{21}^{(d_2)} \right\|^2 + \left\| \mathbf{e}_{24}^{(d_2)} \right\|^2 + \left\| \mathbf{e}_{25}^{(d_2)} \right\|^2 \right). \quad (14)$$

The observation of $L_1$ in $D_1$ does not depend on other sensor positions and landmarks. The partial derivatives of $\mathbf{q_i}$ and landmarks $\mathbf{l}_2, \cdots, \mathbf{l}_5$ are zero. For $\psi = (\psi_{i1}, \psi_{i2}, \mathbf{l}_1, \cdots, \mathbf{l}_2)^T$,

$$\frac{\partial \mathbf{e}_{11}}{\partial \mathbf{q}} = \left( \frac{\partial \mathbf{e}_{11}}{\partial \mathbf{q_{i1}}}, \mathbf{0}_{2\times6}, \frac{\partial \mathbf{e}_{11}}{\partial \mathbf{l}_1}, \mathbf{0}_{2\times3}, \mathbf{0}_{2\times3}, \mathbf{0}_{2\times3}, \mathbf{0}_{2\times3} \right) \quad (15)$$

As the UAV position dimension is smaller than the landmark dimension, the matrix block corresponding to $D_1$ is wider than the matrix block corresponding to $L_1$. If there is at least one common observation between the sensor $D_1$ and the sensor $D_2$, then the corresponding matrix element is set to 1. Similarly, a matrix element $S_{34}$ being zero indicates there is no common observation between $D_3$ and $D_4$ [61], in the connections between all UAV trajectories for landmarks optimization and UAV trajectory [47]. The number of landmarks is much greater than the sensor nodes, and a landmark is often associated with hundreds of feature-points [144]. If the sparsity is used, tens of thousands of feature points can be represented in a compact form without limiting the application scenarios of SLAM. The ground-truth trajectory is composed of multiple layers of different sizes that generate odometry edges from $n - 1$ to $n$ [87]. The edges between layers add observation noise to each edge and reset the initial value of the reference position according to the noisy odometry data [159]. The accumulated error from the initial values of the noisy data approximates the true UAV position $q_x$, $q_y$, $q_z$ to respond to the image data for real-time

SLAM, UAV tracking and landmark mapping [160]. The UAV sensors capture several scenes. The environment and landmark information may be impacted by the intrinsic distortion parameters of sensors [72]. Each sensor has a total of 6-dimensional frame-of-discernment (FoD) to observe landmark nodes and capture point clouds [26,44,47,80]. High dimensionality of landmarks results in computation that cannot be processed in real-time.

The landmark $\mathbf{l}$ is a 3D point-cloud, and the observation data are the pixel coordinates $\mathbb{R}^x \times \mathbb{R}^y \times \mathbb{R}^z \overset{\Delta}{=} [q_x, q_y, q_z]^n$ [27]. If $\mathbf{d}_{ij}$ is the data generated by observing landmark $\mathbf{l}_j$ at the UAV trajectory $\mathbf{q}_n$, then the overall cost function may be represented as:

$$\frac{1}{2}\sum_{i=1}^{m}\sum_{j=1}^{n}\|\mathbf{e}_{ij}\|^2 = \frac{1}{2}\sum_{i=1}^{m}\sum_{j=1}^{n}\|\mathbf{d}_{ij} - d(\mathbf{q}_n, \mathbf{l}_j)\|^2 \tag{16}$$

Equation (16) is equivalent to finding landmarks and the descending direction $\Delta\mathbf{e}_{ij}$, to optimize the objective function [86]. As error is accumulated,

$$\frac{1}{2}\left\|d(\mathbf{e}_{ij} + \Delta\mathbf{e}_{ij})\right\|^2 \approx \frac{1}{2}\sum_{i=1}^{m}\sum_{j=1}^{n}\left\|\mathbf{e}_{ij} + \boldsymbol{\psi}_n\Delta\mathbf{q}_i + \boldsymbol{\psi}_{ij}\Delta\mathbf{l}_j\right\|^2 \tag{17}$$

where $\boldsymbol{\psi}_{ij}$ is the $i$-th position that captures the $j$-th landmark. The $\mathcal{TF}$ timeframes are from the current timestep and remove the earlier ones [151]. When the UAV is paused, the images are from a single point. The features that are observed in the current timeframe correspond to the mean and covariance [140,145,151]. If there are $M$ landmark points $l_1, \ldots, l_M$, they form a local map together with the $\mathcal{TF}$ timeframes [64,142]. The conditional distribution of the UAV positions is $[q_1, q_2, \ldots, q_N \mid l_1, l_1, \ldots, l_M] \sim P([\boldsymbol{\mu}_1, \boldsymbol{\mu}_2, \ldots, \boldsymbol{\mu}_N]^T, \mathbf{C})$ where, $\boldsymbol{\mu}_k$ is the mean of the $\mathcal{TF}$-th timeframe and $\mathbf{C}$ is the covariance matrix over all the timeframes.

*5.3. Adding New Timeframes $\mathcal{TF}$ and Landmarks $\mathcal{L} = [l_1, l_2, \cdots, l_L]$ in UAV Trajectory and Removing Older Ones*

Unused data that are filtered out may affect SLAM accuracy. Considering that $m$ landmarks obey Gaussian distribution mean and variance, a new landmark $l_{N+1}$ adds a variable to the collection of $N + 1$ landmarks [114,161,162]. For $m$ landmarks and $d$ sensor data, since there are usually far more landmarks than sensors, $M \gg N$. The non-zero elements in the $\mathbf{S}$ matrix correspond to the association between the sensor data and the landmark [163]. If there is the same observation made by two sensors, it is referred to as co-visibility. If the $\mathbf{S}$ matrix elements are zero, it indicates that the two sensors do not share landmark observation, indicating no common observations between the corresponding sensor variables $D_1$, $D_2$, $D_3$, and $D_4$. Furthermore, $\psi_1$ views the landmarks from $l_1$ to $l_4$ denoted by $P(\psi_2, \psi_3, \psi_4, l_1, \ldots, l_4|q_1)$. This notation indicates that the landmarks have one more constraint saying where they should be if $q_1$ is set to the current value of conditional distribution [25]. A priori constraint localizes the search space map where the landmark should be at a given time. Landmarks from $l_1$ to $l_4$ may fall into the categories listed below:

1.  The landmark is only observed in $l_1$. In this scenario,

$$P(q_1, \ldots, q_4, l_1, \ldots l_5) = P(q_2, \ldots, q_4, l_1, \ldots, l_5 \mid q_1) \underbrace{p(q_1)}_{\text{discarded}} . \tag{18}$$

2.  Add a new timeframe $\mathcal{TF}$ into the window as well as its corresponding landmarks.
3.  Delete an old timeframe $\mathcal{TF}$ in the window which may also delete the landmarks it observes.
4.  The landmark is seen in $l_2$–$l_4$, but may not be seen in the future if UAV avoids loop closure. To track the missing feature points, this landmark needs a priori information of the future pose estimation.

5. The landmark is seen in $l_2$–$l_4$ and may be seen in the future. This landmark will be estimated later [75]. The observation of this landmark by $q_1$ can be discarded if $q_1$ did not see it.

From the current estimated value, the conditional probability of other state variables and the observed landmark points generate a priori information about where these landmarks should be, which affects the estimated value of the landmark points [164]. For large-scale mapping systems, after several observations, the spatial positions of the landmarks converge to a fixed value and remain unchanged. An occluded landmark is considered as an outlier. The convergent landmarks are fixed to the feature points after a few iterations as constraints of pose estimation to optimize their position [165].

## 6. Open Issues and Future Research Directions in UAV-SLAM

### 6.1. Open Issues

- The number of landmark points to be scanned while a UAV navigates a trajectory range in the thousands. The level of detail to be captured and the required point-cloud density imposes constraints on the resulting data for targetting landmarks distributed over a large area. For example, to capture building facades and utility lines, the usable range of a UAV sensor varies with distance and may not capture dense and detailed data representation [26].
- Loop closure impacts both the localization and map building that enable a UAV to identify the scenes it has visited before. Sensors set additional constraints on the application environment when detecting similarities between images. The accumulated error can be reduced by calculating similarities of images and reliable loop detection eliminates cumulative errors for globally consistent trajectories and maps [80].
- Sensor fusion and navigation for UAVs are usually of large dimensions because all landmark variables are considered by sensors between consecutive positions and landmarks. In visual SLAM, a single image contains hundreds of feature points, which greatly increases the feature-set dimensions. Such a feature matrix is of $O(n^3)$ complexity, which is very expensive in computation. UAV-SLAM requires mechanisms to limit the problem scale to maintain real-time calculations as a UAV navigates a trajectory. As the UAV computing power is limited, calculating SLAM estimates at every moment limits the calculation time for landmarks as the iterations cannot exceed a certain upper bound. In real-time UAV-SLAM, the computational time must not exceed a few milliseconds [134].
- The complexity increases when, instead of using images, the landmark data are extracted from continuous video at regular timeframes. With limited computation, the timeframes may be used only for localization and do not contribute to mapping or vice versa. The number of timeframes increases as the scale of the map grows, limiting SLAM accuracy in real-time computing. If there are $N$ timeframes in a window, and their positions $\psi_1, \dots, \psi_N$, are known in the vector space, then the previous timeframe estimates must remain unchanged during optimization, discarding the variables outside the window [74].

### 6.2. Future Research Trends

In general, UAV-SLAM needs to be robust to perception failures. In UAV environments, there needs to be a trade-off between the sensor's fusion capabilities and the on-board computing power. Visual SLAM can benefit from illumination invariant template matching [126,127]. Some future avenues for improving UAV-SLAM are discussed as follows:

### 6.2.1. UAV Data Fusion for Static and Dynamic SLAM

Compared to SLAM under static conditions, dynamic SLAM divides landmark data into static and dynamic categories for different UAV trajectories. SLAM architectures can be improved to leverage each feature to provide robust localization for UAVs that operate in complex dynamic environments. Additionally, to meet the demands of some high-level

applications such as medical equipment delivery, dynamic SLAM can be integrated with multiple object tracking [99].

### 6.2.2. Deep Learning Based Data Fusion and SLAM

Neural networks have been shown to complement the end-to-end learning requirements in sensor and environment modeling. A probable research avenue is to apply transfer learning between domains that could lead to novel ways to interpret scenes for real-time object detection. In UAVs, a single-UAV intelligence seems insufficient for complex SLAM scenarios and diverse navigation trajectories. Furthermore, C-SLAM may utilize local learning, inter-UAV communication, and data fusion for an adaptive global consensus based on multi-UAV intelligence. In conventional Bayesian SLAM, intermittent and unreliable communication among UAVs may be an impediment to accuracy. However, in deep learning based methods, techniques such as generative adversarial networks (GANs) can be used to reduce the impact of lost communication and missing data. Deep learning based end-to-end localization and mapping are seen as potential alternatives or complementary solutions for absolute UAV localization, instead of SLAM [7].

### 6.2.3. UAV Imagery Impacted by Altitude and Illumination Conditions

In visual SLAM, the computing power is limited by drift, sensor noise $\mathcal{N}_{1:n}$ and accumulated error over time. The UAV trajectory may be represented as an area enclosed through data visualization, localization and mapping. Mapping data can also assist in UAV collision avoidance through state–space estimation to interpret if a given UAV trajectory and associated landmark configurations may lead to a collision between the UAV and the landmark, or not. Furthermore, how UAV sensors perform in the presence of clouds, fog, mist and other climatic conditions for VO is open territory for research [126,127].

### 6.2.4. Opportunities for Improving the Statistical Dependence between Sensor Data Metrics and SLAM

UAV mapping based on homogeneous point-cloud matrices obtained from LiDAR points may only represent the UAV coordinate in a specific timeframe. The landmark data may be classified based on simplicity and due to real-time SLAM computation requirements, LiDAR may not classify certain landmarks that have dimensions that are too big or too small. Furthermore, some landmarks may be approximated by point sequences of up to $n$ points grouped together, where each group is represented by the distance between the first and the last point in the sequence. To ensure safe UAV path planning, the dimensions of all objects in the landmark scene need to be retrieved in the reference map [4,5,47].

### 6.2.5. Accurate and Precise Geo-Referenceing of Landmark Data Using Google Maps

For UAV localization, data related to multiple UAV trajectories and variations in reference maps is needed. The error in trajectory or terrain classification along longitude–latitude–altitude pairs may increase with image depth. From Equation (7), the previously collected data are assumed to be accurately and precisely geo-referenced. In SLAM and VO, UAV localization depends on previously collected data. When these data are aerial imagery or photogrammetry, then accuracy depends on the image processing algorithm. Geo-referencing UAV aerial imagery and photogrammetry data using on board GPS/GNSS will be future work. The 2-DoF and 4-DoF sensor data are a subset of the 6-DoF $= \{x, y, z, \theta_1, \theta_2, \theta_3, \}$. Lack of correlation between several landmarks in the observed scene can arise due to different sensor viewpoints. Using Google maps for landmark matching may add data-dimensions to enhance accuracy and reduce the impact of scale change, blurred data, noise, drift, sensor rotation, occlusion etc. [4,5,47].

### 6.2.6. Reduction in Feature Space for Faster SLAM

By establishing point-by-point correspondence between a number of images, objects in all images are considered to be geometrically aligned. Combining features from the

original images to form a single enhanced image needs image fusion techniques divided into spatial domain and transform domains. A map is constructed as the autonomous UAV explores an environment. SLAM takes advantage of multiple previous observations to jointly estimate landmark positions and the UAV trajectory. The current timeframe $\mathcal{TF}_n$ observed by the UAV is compared to the previous timeframe $\mathcal{TF}_{n-1}$ and the difference is analyzed. Based on this timeframe data $\mathcal{TF}$, the current UAV position $\mathbf{q}_n$ is estimated by adding the most recent estimated position $\mathbf{q}_{n-1}$ to previous $\mathbf{q}_{n-2}$ observations. Reduction in feature space at each timeframe can speed up UAV localization and mapping [146,147].

## 7. Conclusions

In this article, we reviewed and studied the recent trends and developments in SLAM and data fusion for object detection and scene perception in UAVs. The analysis of prevailing SLAM architectures, frameworks and models reveals that a combination of data fusion and SLAM can assist in autonomous UAV navigation without having a predefined map of trajectory or ground-based landmark entities. However, a challenge in multimodal sensor fusion is the grouping of certain sensor data together to reduce redundancy and computational complexity. Although data fusion does not eliminate the underlying sources of error, it aims to reduce the impact of errors accumulated due to different sensor resolutions.

The paper discussed that a single-UAV reference map may contain data from a number of sensors under different levels of illumination and environmental conditions. To tackle uncertainty in sensor data, UAV-SLAM is formulated as a state estimation problem where the current UAV location is used as a reference for future localization and to simultaneously create a reference map. This approach requires frequent data comparison to obtain similarity measures, so that mutual information or redundant data in the search space can be removed to reduce the computational cost of the SLAM architecture. Another challenge in conventional SLAM is to derive one single output from different sensor measurements considering different delays and drifts. Sensor fusion platforms offer complementary, redundant, or cooperative data abstraction levels. The findings of the survey revealed that as the raw sensor data is directly provided as an input to the SLAM algorithms, the latent variables in the data may be lost while arriving at a decision. The data fusion process may be made more end-to-end using machine learning techniques. The Bayesian SLAM can be made more intelligent with deep learning techniques that use a feature-in decision-out approach for end-to-end data fusion. This would enable UAV-SLAM to obtain a set of sensor data with diverse characteristics as an input, and return a UAV trajectory, position and target landmark map as an output.

**Author Contributions:** Conceptualization, A.G. and X.F.; methodology, A.G.; writing—original draft preparation, A.G.; writing—review and editing, X.F.; supervision, X.F.; funding acquisition, X.F. All authors have read and agreed to the published version of the manuscript.

**Funding:** This research received no external funding.

**Institutional Review Board Statement:** Not applicable.

**Informed Consent Statement:** Not applicable.

**Data Availability Statement:** Not applicable.

**Conflicts of Interest:** The authors declare no conflict of interest.

## Abbreviations

The following abbreviations are used in this manuscript:

| | |
|---|---|
| 1D | One Dimensional |
| 2D | Two Dimensional |
| 3D | Three Dimensional |
| 5G | Fifth Generation (Communication networks) |

| | |
|---|---|
| 6-DoF | Six Degrees of Freedom |
| C-SLAM | Collaborative Simultaneous Localization and Mapping |
| DARPA | Defense Advanced Research Projects Agency |
| DGPS | Differential GPS |
| EKF | Extended Kalman Filter |
| FoD | Frame of Discernment |
| GNSS | Global Navigation Satellite System |
| GPS | Global Positioning System |
| ICP | Iterative Closest Point |
| IMU | Inertial Measurement Unit |
| IR | Infra Red |
| KF | Kalman Filter |
| LED | Light Emitting Diode |
| LiDAR | Light Detection and Ranging |
| MAP | Maximum a Posteriori |
| MLE | Maximum Likelihood Estimation |
| NLoS | Non-Line-of-Sight |
| RBPF | Rao Blackwellized Particle Filter |
| RGB-D | Red Green Blue-Depth |
| RPAS | Remotely Piloted Aircraft System |
| RSSI | Received Signal Strength Indicator |
| RTK-GPS | Real-Time Kinematics based Global Positioning System |
| SLAM | Simultaneous Localization and Mapping |
| SLR | Single-Lens Reflex |
| ToA | Time of Arrival |
| UAV | Unmanned Aerial Vehicle |
| VLC | Visible Light Communication |
| VO | Visual Odometry |
| Wi-Fi | Wireless Fidelity (generic term for IEEE 802.11 communication standard) |

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
