# Peer review of "Simultaneous Localization and Mapping (SLAM) and Data Fusion in Unmanned Aerial Vehicles: Recent Advances and Challenges"

_drones, doi:10.3390/drones6040085_

Round 1

Reviewer 1 Report

In this paper, the authors have presented a detailed survey on simultaneous localization and mapping (SLAM) and data fusion techniques for object detection and environmental scene perception in unmanned aerial vehicles (UAVs). The findings are summarized to correlate prevalent and futuristic SLAM and data fusion for UAV navigation and some avenues for further research are discussed., In general, the article is timeless and interesting. The abstract is clear. The references are appropriate. Please consider my detailed comments below.   1. The authors should discuss the motivation and importance of the survey in detail.

2. The advantages of SLAM and data fusion in unmanned aerial vehicles should be highlighted.   3. The literature related to SLAM and data fusion in unmanned aerial vehicles should be discussed. It will be much more useful if the author may provide a table related to this literature.

4. Unmanned Aerial Vehicles (UAVs) have attracted significant attention attributed to their high mobility, low cost, and flexible deployment. The author should discuss the applications of UAVs in various domains.  

Reviewer 2 Report

In this article, the authors give a good overview of the current state and trends in the field of UAV localization and SLAM technologies, but from their own subjective position. The review is made not on the basis of real cases of using real unmanned aerial systems, but in a rather general form, which does not allow starting a substantive dispute on a number of statements at the review stage. In connection with this form of presentation of the material, it is difficult to disagree with the conclusions. In this regard, it seems to me that the article should be assigned not to the Article type, but to the Review type, and in this case it can be published.

Reviewer 3 Report

The manuscript summarizes the SLAM technology. But the discussing points are vague especially in tables 2-4. I could not get original findings from the article.

Reviewer 4 Report

The topic of this submission is spot on for the special edition it is submitted for. The paper discusses advances and challenges to Simultaneous Localization and Mapping (SLAM) and Data Fusion in Unmanned Aerial Vehicles. 

This review is structured as follows:

A) Broad comments: Researchgeneral questions, suggestions, and recommendations.

B) Specific comments: Recommendations and errors need to be fixed.

C) Figures and tables recommendations.

A:

  1. Consider organizing the keywords in alphabetical order. 
  2. Conclusion: there should be a better justification for why the study of recent advances and challenges of SLAM is essential and how it is applied in UAVs. 
  3. It would be better in my view to take UAV applications into a separate section, where you can expand on related aspects of UAVs and SLAM requirements. 
  4. I cannot see the evaluation of current SLAM implementations in robotics and autonomous vehicles and their applicability and scalability to UAVs.
  5. Highlight what is new or different between the related works and compare the previously published papers systematically. It would be amazing to add tables that summarize related works. 
  6. Where is the summary of the findings that correlate prevalent and futuristic SLAM and data fusion for UAV navigation?
  7. Some insightful discussion of how such open issues are affecting the development and causing challenges to UAV-SLAM is needed.
  8. The future research discussed would benefit from a couple of references. 

B:

Consider revising the paper for any grammatical and flow issues. 

Examples:

  1. Fixing section format (numbering) in the last paragraph in the introduction.
  2. Make sure that the abbreviations are mentioned the first time they are used. For example, line 726 à Visual odometry (VO) abbreviation is not the first time used. 

C:

  1. Tables 2,3, and 4: need to be concise, clearer, and more organized. 

Reviewer 5 Report

SLAM and data fusion are trending topics in UAVs, but there is not a clear contribution in this paper. This document seems more like a lecture on data fusion and SLAM that a research review paper. 

The authors do not justify why deep learning approaches are excluded in the review when they themselves recognize that they are state of the art.

There is an excess of references (almost every sentence has one or two cites!), but the relevant ones are missing. The paper does not mention other relevant surveys and does not state in which way their work differs from existing ones. 

The authors often generalize unjustifiably. UAVs not always follow a non-deterministic path and SLAM is not always necessary. See for example precision agriculture applications, where the UAV will follow a predefined scan pattern and GNSS precision is more than enough.

The introduction must be restructured. Subsection title "1.1 UAV applications", for example, is not appropriate. First, a summary of well-known UAV applications is not required. It would be interesting if the paper focuses on which ones really require or can benefit from SLAM. Second, from line 102, the text is not related to UAVs application, but it is still inside the subsection.

In section 2 line 172, "longitude-latitude pairs corresponding to the geographical coordinates" should consider altitude.

Line 221, LIDAR does not capture images!

Round 2

Reviewer 3 Report

I could not find information of accuracy or precision in the article. 

Reviewer 4 Report

Dear authors of the manuscript,

You are most welcome, and excellent job revising the previous points. 

I still have trouble understanding tables 3-5. They are still can not easily be understood and can lead to confusion. Please consider revising them and adding explanations in the text.

Reviewer 5 Report

The paper has been significantly improved, in particular with the addition of the new tables. 

The main drawback is still the lack of references to other UAV-SLAM surveys. There are too many (unnecessary) references, but the ones from previous related works are missing. 
